# Seaweed Phenolics: From Extraction to Applications

**DOI:** 10.3390/md18080384

**Published:** 2020-07-24

**Authors:** João Cotas, Adriana Leandro, Pedro Monteiro, Diana Pacheco, Artur Figueirinha, Ana M. M. Gonçalves, Gabriela Jorge da Silva, Leonel Pereira

**Affiliations:** 1MARE-Marine and Environmental Sciences Centre, Department of Life Sciences, University of Coimbra, 3001-456 Coimbra, Portugal; jcotas@gmail.com (J.C.); adrianaleandro94@hotmail.com (A.L.); dianampacheco96@gmail.com (D.P.); amgoncalves@uc.pt (A.M.M.G.); 2Faculty of Pharmacy and Center for Neurosciences and Cell Biology, Health Sciences Campus, University of Coimbra, Azinhaga de Santa Comba, 3000-548 Coimbra, Portugal; pmdmonteiro1992@gmail.com (P.M.); gjsilva@ci.uc.pt (G.J.d.S.); 3LAQV, REQUIMTE, Faculty of Pharmacy of the University of Coimbra, University of Coimbra, Azinhaga de Santa Comba, 3000-548 Coimbra, Portugal; amfigueirinha@gmail.com; 4Faculty of Pharmacy of University of Coimbra, University of Coimbra, 3000-548 Coimbra, Portugal; 5Department of Biology and CESAM, University of Aveiro, 3810-193 Aveiro, Portugal

**Keywords:** seaweed polyphenolics, polyphenolics extractions, phlorotannins, bromophenols, flavonoids, phenolic terpenoids, polyphenolics bioactivities

## Abstract

Seaweeds have attracted high interest in recent years due to their chemical and bioactive properties to find new molecules with valuable applications for humankind. Phenolic compounds are the group of metabolites with the most structural variation and the highest content in seaweeds. The most researched seaweed polyphenol class is the phlorotannins, which are specifically synthesized by brown seaweeds, but there are other polyphenolic compounds, such as bromophenols, flavonoids, phenolic terpenoids, and mycosporine-like amino acids. The compounds already discovered and characterized demonstrate a full range of bioactivities and potential future applications in various industrial sectors. This review focuses on the extraction, purification, and future applications of seaweed phenolic compounds based on the bioactive properties described in the literature. It also intends to provide a comprehensive insight into the phenolic compounds in seaweed.

## 1. Introduction

Recently, organisms living in aquatic habitats have been gaining more interest as the target of studies by numerous scientific groups that have mainly studied their pharmaceutical and biomedical properties, such as the antioxidant, anti-inflammatory, anti-fungal, anti-bacterial, and neuroprotective activity of their large diversity of bioactive compounds [1,2,3,4,5].

Seaweeds are considered the sea vegetables and the basis of life in the aquatic habitats, and they have been employed as fertilizer, human food, and animal feed from ancient to modern times [6,7]. Seaweeds are classified into three main groups, Chlorophyta (green seaweeds), Rhodophyta (red seaweeds), and Phaeophyceae (brown seaweeds), according to their composition of pigments, and their composition of metabolites differs vastly [8].

Seaweeds are commonly exposed to harsh environmental conditions, and the damage effects on them are not visible; consequently, the seaweed produces a large range of metabolites (xanthophylls, tocopherols, and polysaccharides) to protect from abiotic and biotic factors, such as herbivory and sea mechanical aggression [9,10,11,12,13,14,15]. Note that seaweed metabolites content and diversity is subject to abiotic and biotic factors, such as species, life stage, size, age, reproductive status, location, depth, nutrient enrichment, salinity, light intensity exposure, ultraviolet radiation, intensity of herbivory, and time of collection; thus, the full exploitation of algal diversity and complexity requires knowledge of environmental impacts and an understanding of biochemical and biological variability [16,17]. Likewise, seaweeds have nutraceutical and pharmaceutical compounds, such as phenols and chlorophylls [6].

Phenolic compounds are found in land plants and in seaweeds [18,19]. Polyphenols synthesized by seaweeds, as one of the largest and most widely distributed groups of seaweed phytochemicals, have gained special attention due to their pharmacological activity and array of health-promoting benefits, as polyphenols play a significant role in the high variety of seaweed biological activities [2,8,14].

Seaweed phenolic compounds are metabolites that are chemically characterized as molecules containing hydroxylated aromatic rings [14,15,20]. These phytochemicals show a wide variety of chemical structures, from simple moieties to high molecular polymers. The biogenetically primary synthetic pathways that produce these phytochemicals are the shikimate or the acetate pathways [21,22,23].

Seaweeds are a valuable source of polyphenolic compounds such as phlorotannins, bromophenols, flavonoids, phenolic terpenoids, and mycosporine-like amino acids. In the brown seaweeds, the phlorotannins are the major polyphenolic class found only in the marine brown seaweeds. On the other hand, the largest proportion of phenolic compounds present in green and red seaweeds are of bromophenols, flavonoids, phenolics acids, phenolic terpenoids, and mycosporine-like amino acids [24,25,26,27]. These molecules are considered secondary metabolites, as they are protective agents that are produced in response to different stimuli and are defense mechanisms of the seaweeds against herbivory and UV radiation [4].

Phenolic compounds are very difficult to isolate quantitatively at an industrial scale due to their structural similarity and tendency to react with other compounds [14]. However, these compounds possess chemical properties that enable their extraction and purification, allowing highly purified extracts to be obtained at a lab scale [28].

Most phenolic compounds possess a broad variety of biological activities, as anti-diabetic, anti-inflammatory, anti-microbial, anti-viral, anti-allergic, anti-diabetic, antioxidant, anti-photoaging, anti-pruritic, hepatoprotective, hypotension, neuroprotective and anticancer properties [14,27,29,30,31,32,33,34,35,36,37,38,39,40]. Most of the bioactivities are related to the interaction of phenolic compounds with proteins (enzymes or cellular receptors) [41].

These wide-range bioactivities make seaweeds candidates for the development of products or ingredients used in industrial applications as pharmaceuticals, cosmetics, functional foods, and even in bioactive food packaging films to maintain the quality of food products [37,39,42,43,44].

Simple phenolic compounds can act as an intermediate in the biosynthesis of many polyphenolic secondary metabolites, being also an essential precursor for the industrial synthesis of many other organic substances. Inclusively, salts of benzoic acid are used as industrial food preservers [15,35].

This review aims to present a comprehensive insight into the seaweed phenolic compounds, providing important information about the current and potential status of these compounds from their origin to extraction and isolation methodology, highlighting the potential activities and commercial applications of these compounds in various industries or the potential to become new products.

## 2. Phenols Found in Seaweeds

Structurally, phenolic molecules are characterized by the presence of an aromatic ring with one or more hydroxyl groups, with structures ranging from simple molecules, such as hidroxycinnamic acids or flavonoids, to more complex polymers, which are characterized by a wide range of molecular sizes (126–650 kDa) [45,46,47,48,49,50,51]. The name “phenol” refers to a substructure with one phenolic hydroxyl group; cathecol and resorcinol (benzenediols) are characterized by two phenolic hydroxyl groups; and pyrogallol and phloroglucinol are characterized by three hydroxyl groups (benzenetriols) [48].

### 2.1. Phenolic Acids

Phenolic acids (PAs) are bioactive compounds involved in several functions, including nutrient absorption, protein synthesis, enzymatic activity, photosynthesis, and allelopathy. These are regularly bound to other molecules, such as simple and/or complex carbohydrates, organic acids, and other bioactive molecules such as flavonoids or terpenoids [52,53,54]. These PAs are formed by a single phenol ring and at least a functional carboxylic acid group, and they are usually classified depending on the number or carbons in the chain, which is bound to the phenolic ring. Accordingly, these phenolic acids are classified as C6-C1 for hydroxybenzoic acid (HBA; with one carbon chain attached to the phenolic ring), C6-C2 for acetophenones and phenylacetic acids (two carbon chains attached to the phenolic ring), and C6-C3 for hydroxycinnamic acids (HCA) (3 carbon chains attached to the phenol ring) [52,53,54].

HBAs includes gallic acid, p-hydroxybenzoic, vanillic, and syringic acids and protocathecins, among others, in which there are variations in the basic structure of the HBA, including the hydroxylation and methoxylation of the aromatic ring [52,53,54]. Although these can be detected as free acids, they occur mainly as conjugates [52,53,54]. For example, gallic acid and its dimer ellagic acid may be esterified with a sugar (usually glucose) to produce hydrolysable tannins [52,53,54].

Hydroxycinnamic acids (HCA) are trans-phenyl-3-propenoicacids, differing in their ring constitution [52]. These HCA derivatives include caffeic (3,4-dihydroxycinnamic), ferulic (3-methoxy-4-hydroxy), sinapic (3,5-dimethoxy-4-hydroxy), and p-coumaric (4-hydroxy) acids, with a wide distribution of these compounds as conjugates, mainly as esters of quinic acid (chlorogenic acids [CGA]) [52,53,54]. Additionally, depending on the identity, number, and position of the acyl residue, this acids can be subdivided in several groups: (1) mono-esters of caffeic, ferulic, and p-coumaric; (2) di-, tri-, and tetra-esters of caffeic acids; (3) mixed di-esters of caffeic–ferulic acid or caffeic–sinapic acids; and (4) mixed esters of caffeic acid with dibasic aliphatic acids, such as oxalic or succinic [52,53,54]. Furthermore, cinnamic acids can condense with molecules other than quinic acid, including rosmaric and malic, with aromatic amino acids and choline, among others [52,53,54].

In seaweeds, there are some studies that have proven the presence of the PAs. However, these studies are scarce and mainly in phenolic characterization without any bioactivities studied [52,53,54,55]. In green seaweeds, coumarins have been identified in species such as Dasycladus vermicularis, as well as some vanilic acid derivatives in the green macroalgae Cladophora socialis [56]. In brown seaweeds, HBAs, rosmarinic acid, and quinic acid derivatives have been characterized in *Ascophyllum nodosum*, *Bifurcaria bifurcata,* and *Fucus vesiculosus* [57]. In addition, Pas has been characterized in the genus *Gracilaria*, as well as benzoic acid, p-hydroxybenzoic acid, salicylic acid, gentisic acid, protocatechuic acid, vanillic acid, gallic acid, and syringic acid [58,59,60].

### 2.2. Phlorotannins

Of all the seaweed phenolic metabolites, the main attention has been focused on tannins due to their interesting bioactive properties [49].

Phlorotannins (Figure 1) can be found within the cell in vesicles called physodes, which are located both in the periphery of the cell and in perinuclear regions, where they are formed [48,49,61]. Phlorotannins are oligomers of phloroglucinol, which is restricted to brown seaweeds, where they exert functions as primary and secondary metabolites [48]. The monomeric unit of phlorotannins, phloroglucinol, is assumed to be formed through the acetate–malonate (polyketide) pathway, in the Golgi apparatus [48,61]. Two molecules of acetyl-CoA, in the presence of carbon dioxide, are converted into malonyl-CoA. The polyketomethylene precursor formed by 3 malonyl-CoA blocks is subjected to a “Claisen type” cyclization reaction, forming a hexacyclic ring system, which is not thermodynamically stable. Then, this molecule undergoes tautomerization, forming a more stable molecule of phloroglucinol [62,63]. The phloroglucinol residues bind through C–C and/or C–O–C residues to form polymeric molecules of phloroglucinol, which results in molecules ranging between 10 and 100 kDa whose heterogeneity is attributed to the variability of structural linkages between phloroglucinol and the hydroxyl groups present [62,63]. As such, phlorotannins can be subdivided into six groups, according to the nature of the structural linkage: (1) phloretols (aryl–ether bonds); (2) fucols (aryl–aryl bonds); (3) fucophloretols (ether or phenyl linage); (4) eckols (dibenzo-1,4-dioxin linkages); (5) fuhalols (*ortho-/para-* arranged ether bridges containing an additional hydroxyl group on one unit), and (6) carmalols (dibenzodioxin moiety) [48,62,63].

Additionally, the binding of monomers to the phloroglucinol ring can take place at different positions within each class, leading to the formation of structural isomers in addition to the conformational isomers [62,63]. As the structural complexity increases, it is necessary to use other criteria for classification. As such, compounds for each class can be classified as linear phlorotannins (in which C–C/C–O–C oxidative couplings have two terminal phloroglucinol residues) or branched phlorotannins, if they are bound to three or more monomers [62,63].

### 2.3. Bromophenols

Bromophenols (BP) (Figure 2) are secondary metabolites with ecological functions, such as chemical defense and deterrence, with studies revealing a wide variety of beneficial ecological activities [33,61]. BP are common to all major algal groups [61]; they were first isolated from the red algae *Neorhodomela larix* (formerly known as *Rhodomela larix*) [64] and thereafter identified and isolated from all taxonomic groups of marine macroalgae, such as red [64,65,66], green [67,68,69], and brown algae [70,71,72,73].

Bromophenols are characterized by the presence of phenolic groups with varying degrees of bromination. Many seaweed species contain haloperoxidases, which are capable of halogenating organic substrates in the presence of halide ions and hydrogen peroxide. The bromoperoxidase activity, isolation, and characterization of haloperoxidases have been demonstrated from seaweed [33,74], but information regarding the biosynthesis of bromophenols is limited [74]. Bromoperoxidases can brominate phenol, resulting in the formation of bromophenols; however, the precursor of such bromophenols is still not established, with some authors suggesting tyrosine as a precursor for such formation [75].

Compared to phlorotannins, less is known about bromophenols due to the limited quantity of these compounds in the seaweeds, with a consequently lower isolation and bioactive characterization. More work is needed to isolate and chemically characterize this group of molecules. Yet, there are already some studies correlating the isolated compound with bioactivities and mechanisms of action [33]. Some of these studies have been conducted with synthetic bromophenols developed from the chemical characterization of the seaweed bromophenols that are mainly identified in red seaweeds [76,77]. The bromophenols are vital for the seaweed and seafood flavor, namely 2-bromophenol, 4-bromophenol, 2,4-dibromophenol, 2,6-dibromophenol, and 2,4,6-tribromophenol [72].

### 2.4. Flavonoids

Flavonoids (Figure 3) are phenolic compounds structurally characterized by heterocyclic oxygen bound to two aromatic rings, which then can vary according to the degree of hydrogenation [46,51].

These compounds are widely distributed in terrestrial plants, with over 2000 compounds reported, which have been subdivided into major categories such as flavones, flavanol, flavanones, flavonols, anthocyanins, and isoflavones [51]. There are many studies on the flavonolic content in terrestrial plants, but flavonoid content studies in algae are scarce [45]. Some studies report that seaweeds are a rich source of catechins and other flavonoids. Flavonoids such as rutin, quercitin and hesperidin, among others, were detected in several Chlorophyta, Rhodophyta, and Phaeophyceae species [48] and compounds restricted to several macroalgae have been identified, such as hesperidin, kaempferol, catechin, and quercetin [78]. Isoflavones, such as daidzein or genistein, are present in red macroalgae *Chondrus crispus* and *Porphyra*/*Pyropia* spp. and in brown seaweeds, such as *Sargassum muticum* and *Sargassum vulgare*. A high number of flavonoid glycosides have been found in the brown macroalgae *Durvillae antarctica*, *Lessonia spicata,* and *Macrocystis pyrifera* (formerly known as *Macrocystis integrifolia*) [48]. There are already studies that support the presence of flavonoids C-glycosides in the green seaweed *Nitell Hookeri* [79].

It is a little hard to have a full understanding of the bibliography of this phenolic compound’s class in the seaweed, due to the scarce bibliographic support about the seaweed flavonoids, but there are also contradictions between the studies done, where the isolation and characterization of the seaweed’s flavonoids need to be more explored. For example, the work of Yonekura-Sakakibara et al. [80] states that in general, algae (micro and macro-algae) do not present flavonoid content, due to the lack of two primary enzymes for the main flavonoid biosynthesis; however, the genes encoding enzymes of the shikimate pathway were described in algae [80,81]. However, the work of Goiris et al. [82] reveals the presence of flavones, isoflavones, and flavonols in various microalgae evolutionary lineages, such in Rodophyta, Chlorophyta and Ochrophyta.

This also happens in the seaweed flavonoid biosynthesis, where there is the need to research to have a full understanding of the seaweed cell mechanism to produce these specific phenolic compounds. There are some studies that try to explain the flavonoid synthesis by the shikimate–acetate pathway [83]. Yonekura-Sakakibara et al. [80] explain that the p-coumaroyl-CoA, derived from the phenylpropanoid pathway (with tyrosine biosynthesized by the shikimate pathway), and malonyl-CoA, from the acetate–malonate (polyketide) pathway (identical to the phlorotannins pathway), are converted into naringenin chalcone by the chalcone synthase, and spontaneously, the naringenin chalcone is catalyzed by the chalcone isomerase into naringenin. The naringenin is the primary and initial precursor of the all flavonoids synthetized. Afterwards, naringenin is converted into dihydrokaempferol by the flavanone 3-hydroxylase, and there, the enzymatic mechanism transforms this molecule into the diverse flavonoid molecules, such as catechins, flavonols, anthocyanidin, flavones, and others [80].

The quantitative measurement of flavonoid content is a recurrent analysis that is used mainly as a part of biochemical characterization (together with phenolic content analysis) of the seaweeds extracts in the bibliography analyzed. However, there is a lack on the studies about the flavonoid characterization, isolation, and their specific bioactivity analysis, in the bibliography analyzed. There are only suppositions by the high content of flavonoids present in the sample analyzed and the bioactivity analyzed [35,78,84,85,86,87]. From the recent studies using various seaweeds species, the data obtained support the presence of the flavonoid compounds, with various extraction techniques, and these extracts have antioxidant and radical scavenging activity positively correlated with flavonoid concentration [12,88,89]. However, the study of Abirami and Kowsalya [90] did not detect flavonoids in *Ulva lactuca* (Chlorophyta) and *Kappaphycus alvarezii* (Rhodophyta). The detection of flavonoids and respective differences can be derived by the geographical location, time of harvest/collection, season of collection, and the methodology employed in the laboratory [12]. The study from Yoshie et al. [91] reveals the presence of catechin, epicatechin, epigallocatechin, catechin gallate, epicatechin gallate, or epigallocatechin gallate in *Acetabularia ryukyuensis*, *Ecklonia bicyclis*, *Padina arborescens*, *Padina minor*, *Neopyropia yezoensis* (as *Porphyra yezoensis*), *Gelidium elegans,* and *Portieria hornemannii* (as *Chondrococcus hornemannii*). However, this work did not detect flavonoids in *Undaria pinnatifida*, *Monostroma nitidum*, *Caulerpa serrulata*, *Caulerpa racemosa*, *Valonia macrophysa*, *Chondrus verrucosus,* and *Actinotrichia fragilis*.

### 2.5. Phenolic Terpenoids

Phenolic terpenoids (Figure 4) have been detected and characterized in brown and red seaweeds [61]. Brown seaweed phenolic terpenoids have been mainly characterized as meroditerpenoids, which are divided in plastoquinones, chromanols and chromenes, and these are found almost solely in the Sargassaceae. These meroditerpenoids consist of a polyprenyl chain bound to a hydroquinone ring moiety [92].

Diterpenes and sesquiterpenes were identified and isolated in Rhodomelaceae, and a macrolide formation under secondary cyclization was reported for the red seaweed *Callophycus serratus* (bromophycolides) [93].

Even though these compounds have been identified and isolated, the pathways for their formation have yet to be identified, as there is no evidence that these compounds follow the same biosynthesis pathway as other terpenes and terpenoids. Meroditerpenoids are partially formed from mevalonic acid pathways [94], but further studies on biosynthesis pathways should be pursued.

### 2.6. Mycosporine-Like Aminoacids (MAA)

Mycosporine-like amino acids (MAA) (Figure 5) are a group of UV-absorbing compounds that are present in a wide range of aquatic organisms, whose main function is to reduce UV-induced cellular damage [61,95,96,97,98]. These compounds were first identified in fungi, with a role in UV-induced sporulation. Thereafter, a wide range of MAAs have been found in a diverse variety of aquatic organisms, including Cyanobacteria, micro-, and macro-algae [96]. These compounds were detected in *Rhodophyta* spp., and there is still some debate on the detection of MAAs in seaweeds belonging to green and brown seaweed species [61].

Geographically, these compounds are found ubiquitously, occurring in a wide range of environments. Intracellularly, these compounds are found distributed in the cell cytoplasm [96].

These compounds are water soluble, with low molecular weights (<400 Da). The chemical structure is based on a ciclehexenone or cyclohexenine ring, with amino acid substituents. The conjugated bonds within the molecule result in broadband absorptions of different wavelengths, according to the substituents in the chemical structure. Evidence suggests that most of the MAAs are synthesized via the Shikimate pathway [96,97].

### 2.7. Non-Typical Phenolic Compounds

Non-typical phenolic compounds (Figure 6) have been characterized in macro-algal species [61]. Colpol was found in the brown seaweed *Colpomenia sinuosa* [99]. Phenylpropanoid derivatives, such as tichocarpols, were identified in the red macroalgae *Tichocarpus crinitus* [100]. Lignin, a polymerized hydroxicinnamyl alcohol, commonly identified and thought to be restricted to vascular plants, was also identified in red seaweed *Calliarthron cheilosporioides* [101].

## 3. Phenolic Compounds Extractions and Purification Methodologies

For seaweed phenolic compounds investigation, from the pre-treatment to their characterization, there are many methodologies that could be employed (Figure 7). Initially, it is necessary to select the target seaweed species and then locate and identify the compounds to be extracted—intracellular or extracellular—in order to define the strategies to perform the extraction, isolation, and assessment of the phenolic compound bioavailability and bioactivities [36,42,102,103,104,105]. To develop bio-based products with seaweed phenolic compounds, it is pivotal to develop practical and efficient analytical methods.

### 3.1. Pre-Treatment

Seaweed pre-treatment is recommended, such as a washing step to remove stones, sand, epiphytes, or other impurities. Therefore, the algal biomass could be used fresh, dried—air drying or at 30–40 °C with aeration during 3–5 days—or freeze drying [106]. Freeze-dried is a better option because it guarantees the integrity of the biomolecules and allows better extraction yields [107].

Furthermore, a milling or grinding process is suggested to reduce the particle size, which is going to increase the exposure area between the seaweed biomass and the solvent used for extraction [108]. This will consequently increase the extraction yield.

Usually, a pre-extraction process is required to avoid the co-extraction of pigments or fatty acids [48] with low polar solvents—*n*-hexane [109], *n*-hexane:acetone [110], *n*-hexane:ethyl acetate [111], or dichloromethane [112]—which have been shown to be effective to extract phenolic compounds.

For example, an efficient pre-treatment using acetone:water (7:3) was applied to the brown seaweed *Fucus vesiculosus* before the extraction of phlorotannins [113].

### 3.2. Extraction

The further step is to select an extraction method, since these methodologies are widely variable.

Traditional extraction techniques include Soxhlet, solid–liquid, and liquid–liquid extractions. Commonly, in the cited methodologies, organic solvents are used (e.g., hexane, petroleum ether, cyclohexane, ethanol, methanol, acetone, benzene, dichloromethane, ethyl acetate, chloroform). Nevertheless, nowadays, the solvent applied in the extraction methods should be non-toxic and low cost [114]. From an industrial point of view, ethanol is preferred as a solvent for the extraction due to its lower cost [106].

Maceration is a classical method in which the compounds are extracted by submerging the seaweed biomass in an appropriate solvent/solvent mixture [115].

The classical Soxhlet extraction method provides several advantages because it is a continuous process, the solvent can be recycled, and it is less time and solvent consuming than maceration and percolation techniques [116]. However, the extract is constantly being heated at the boiling point of the solvent, and this can damage thermolabile compounds and affect further analysis [45]. Nevertheless, the mentioned extraction methods are not efficient and environmental friendly, due to the high quantities of organic solvent required [23].

With technological advances, these methods have evolved over time to improve the extraction efficiency and sustainability. Currently, ultrasounds and microwave-assisted extraction are low-cost technologies that are feasible at a large scale [117]. These techniques cause a physical effect that leads to cellular membrane disruption and facilitates the liberation of the target compounds [118].

Ultrasound-assisted extraction uses acoustic cavitation to disrupt the cell walls, leading to the reduction of size particle and consequently enhancing the contact between the solvent and the target compound [119].

Still, microwave-assisted extraction involves the utilization of microwave radiation to heat solvents in contact with a sample. Since the algal cell wall is highly susceptible to microwave irradiation, it is reported the rapid internal heating causes cell disruption, leading to the release of target compounds to the cold solvent [120].

Pressurized fluids can also be applied as extraction agents that lead to the development of several techniques, such as subcritical water extraction, supercritical fluid extraction, and accelerated solvent extraction [114].

The correct selection of the extraction solvent, temperature, pressure, static time, and number of cycles are variables that influence directly the total phenolic yield and rate [121]. Further research shows that the total phenol yield increases with the subcritical water extraction method (or pressurized hot water extraction) under higher temperatures; temperatures of 100 °C and 200 °C were tested [122]. This phenomenon could be explained by the increase of mass transfer due to the higher solubility of the cellular membrane, which is caused by the increment of temperature [123]. In this study, the cited trend was verified in all the seaweeds under study, namely *Sargassum vulgare*, *Sargassum muticum*, *Porphyra/Pyropia* spp., *Undaria pinnatifida,* and *Halopithys incurva* [122]. However, the increase of the temperature could also lead to the polymerization or the oxidation of some phenolic substances.

The supercritical liquid is characterized by being environmentally friendly, because it uses supercritical carbon dioxide instead of an organic solvent. This methodology is currently the most employed technique for phenolic compounds extraction, and it is defined by the supercritical state, which occurs when a fluid is above critical temperature and pressure—for example, when it is between the state transition (e.g., gaseous and liquid state) [124]. This extraction is commonly performed at low temperatures, so it is suitable for compounds susceptible to high temperatures. This methodology presents several advantages, because supercritical fluids have a lower viscosity and higher diffusion rate than liquid solvents. For this reason, the mass transfer is faster, and the extraction is more efficient [124]. In opposition, as this method involves carbon dioxide, this technique is only applicable to compounds that are lipid soluble and with low polarity.

Relative to accelerated solvent extraction, it also enhances the extraction speed, has low solvent requirements, and is possible to achieve the highest phenolic compounds extraction yield in comparison with conventional approaches. Thus, it is not suitable for bulk extraction [114].

Extraction using enzymes is not feasible at an industrial scale due to the high cost of the necessary enzymes. Nevertheless, it retains several advantages due to the high selectivity of the enzymatic extraction methods [114].

Enzymes can also be applied to promote cell wall disruption, and extraction using enzymes is an advantageous technique due to the selectivity of the degraded compounds, which is important for fragile and unstable substances. In some cases, enzymes are capable of converting insoluble compounds in water into water-soluble ones [25]. For instance, an enzymatic hydrolysate rich in polyphenols was extracted from the brown seaweed *Ecklonia cava*, in which it obtained a polyphenol yield of 20% with the enzyme Celluclast (Novo Nordisk, Bagsvaerd, Denmark) [125].

Usually, after the extraction process, seaweed extracts are concentrated using a rotary evaporator [58].

### 3.3. Purification, Quantification, and Characterization

Following the extraction process, it is necessary to proceed to the isolation and quantification of the target phenolic compound. Several methodologies could be applied, according to the typology of the compound to be isolated.

Generally, the analysis of phenolic compounds is affected by their source, the extraction and purification techniques employed, the sample particle size, the storage conditions, and the presence of interfering substances in extracts such as fatty acids or pigments [22].

Classically, the quantification of phenolic content is performed by colorimetric methods, namely Folin–Ciocalteu, Folin–Denis, or Prussian blue assays [14]. The most applied assay for phenolic compounds assessment is Folin–Ciocalteu—the redox reaction with the reagent Folin–Ciocalteu allows the spectrophotometric quantification assessment of phenolic compounds. However, the disadvantage of this technique is in the interference of non-phenolic reducing substances [21].

Nowadays, the isolation of phenolic compounds is made through preparative chromatography techniques, namely column chromatography, high-pressure liquid chromatography (HPLC), or thin-layer chromatography (TLC). Nevertheless, these chromatographic techniques have evolved in order to be also used for the separation, isolation, purification, identification, and quantification of distinct phenolic compounds [14].

HPLC, coupled with appropriate detectors, is a very efficient automated analytical methodology that allows the separation, purification, and characterization of a wide range of chemical samples [126]. It presents several advantages because it is a quick method, it requires a low amount of extract sample, and the equipment is easy to operate [48]. Moreover, further research reports that by the HPLC technique, it was possible to identify and quantify nine phenolic compounds (gallic acid, 4-hydroxybenzoic acid, catechin hydrate, epicatechin, catechin gallate, epicatechin gallate, epigallocatechin, epigallocatechin gallate, and pyrocatechol) in brown edible seaweeds (Phaeophyceae)—*Eisenia bicyclis* (formerly known as *Eisenia arborea* f. *bicyclis*), *Sargassum fusiforme* (formerly known as *Hizikia fusiformis*), *Saccharina japonica* (formerly known as *Laminaria japonica*), *Undaria pinnatifida*—and in red edible seaweeds, (Rhodophyta)—*Palmaria palmata* and *Pyropia tenera* (formerly known as *Porphyra tenera*) [103].

The reversed-phase liquid chromatography (RP-HPLC), in which the analysis requires a non-polar stationary phase and a polar hydro-organic mobile phase [126], increases the retention with the hydrophobicity of the solutes, the hydrophobicity of the stationary phase, and the polarity of the mobile phase [127,128]. Thus, the separation is accomplished through the partitioning process and the adsorption of the compounds [129]. Still, the identification and quantification of phlorotannins is usually performed by RP-HPLC with methanol/acetonitrile and water (buffer) solvent combinations and the detection in the UV range of the spectrum [16]. For instance, in a study conducted with the red seaweed *Rhodomela confervoides*, the RP-HPLC method provided bromophenols’ identification and characterization (3-bromo-4, 5-dihydroxy benzoic acid methyl ester and 3-bromo-4,5-dihydroxy- benzaldehyde) [130].

Alternatively, thin-layer chromatography is a methodology for compound identification and isolation in which the stationary phase is an adsorbent layer of fine particles. Overall, the layer is placed on a closed chamber, and the extract sample is applied on the lower side of the layer. Inside of this chamber is the mobile phase, which is characterized by a mixture of solvents. Then, the distance covered is marked for the calculus of the retention factor (Rf), enabling the compound identification [131]. In fact, using this method in a dichloromethane/methanol/water (65:35:10, *v*/*v*/*v*) solvent system, researchers isolated phlorotannins (phlorofucofuroeckol, dieckol, and dioxinodehydroeckol) from *Ecklonia stolonifera* (Phaeophyceae) [132].

The coupling of liquid or gas chromatography with mass spectrometry also enabled the characterization of phenolic compounds [62,133]. Liquid chromatography-mass spectrometry (LC-MS) allowed the analysis of phlorotannins with different degrees of polymerization, from three brown seaweeds—*Durvillaea antarctica*, *Lessonia spicata,* and *Macrocystis integrifolia* (currently known as *Macrocystis pyrifera*) [134]. For instance, gas chromatography-mass spectrometry (GC-MS) allowed the identification of coumarin and flavones on crude extracts of *Padina tetrastromatica* (Phaeophyceae) [135]. However, this methodology is not appropriate for non-volatile compounds.

Matrix-assisted laser desorption/ionization time-of-flight mass spectrometry (MALDI-TOF-MS) is also a technique that provides the identification and structural characterization of biomolecules [136]. Previous research used this technique in order to detect the presence of phloroglucinol derived from the brown seaweed *Sargassum wightii* [137]. Merged with the previous technique, ultrahigh performance liquid chromatography–electrospray ionization tandem mass spectrometry (UHPLC-ESI-MS) provides relevant information for phenolic compounds characterization relative to their size and isomeric variations [48]. Through this technique, a team of researchers was able to identity and characterize 22 phlorotannins from *Fucus* spp. [138].

Recently, quantitative nuclear magnet resonance (qNMR) has shown the efficiency for metabolites identification and quantification [139,140]. In general, NMR spectrum is derived from the measurement of Fourier transformation signals and translated to radio-frequency impulses. Thus, in comparison with other spectroscopy methods, NMR has a lower mass sensitivity [141].

This method was applied after optimized accelerated solvent extraction, and it was possible to observe the phenolic profile of *Ulva intestinalis* (Chlorophyta) [142].

## 4. Seaweed Phenolic Compounds and their Bioactivities

The characterization of some phenolic extracts showed interesting results (Figure 8). The correlation between the specific compound and bioactivity potential is commonly achieved (Figure 9); however, there are some phenolic-enriched extracts that have interesting properties but have not been chemically characterized. In this topic, we describe the compounds isolated and their multi-role activity, focusing on recent studies, so there is a long road toward the development of a final product or solution.

### 4.1. Green Seaweeds

Bromophenols and flavonoids of green seaweeds (Figure 10) have antioxidant activities. In fact, species such as *Ulva clathrata*, *Ulva compressa* (formerly known as *Enteromorpha compressa*), *Ulva intestinalis*, *Ulva linza*, *Ulva flexuosa*, *Ulva australis* (formerly known as *Ulva pertusa*), *Capsosiphon fulvescens*, and *Chaetomorpha moniligera* were tested and proven to have high radical scavenging activities [143,144]. These findings will allow the development of new products in drug, cosmetics, or food industries.

Furthermore, the phenolic fraction of *Ulva clathrata* has an anti-tumoral effect [145], and this was also verified in *Ulva flexuosa* species [146]. The phenolic extract of the last not only showed cytotoxicity against breast ductal carcinoma cell line, but also antibacterial activity. This proves the interest of phenol of green seaweeds for human health.

In more recent studies, the *C. socialis* phenolic compounds, such as 2,3,8,9-tetrahydroxybenzo[c]chromen-6-one, 3,4,3′,4′-tetrahydroxy-1,1′-biphenyl, and cladophorol have been identified with interesting antibacterial activity against methicillin-resistant *Staphylococcus aureus* [147].

#### 4.1.1. Bromophenols

Bromophenols found in green seaweeds revealed interesting properties, such as the 5′-hydroxyisoavrainvilleol isolated from the *Avrainvillea nigricans,* which is cytotoxic to KB cells and demonstrated promising anti-microbial activity [68]. Another promising bromophenol from the same genus, from the species *A. rawsoni*, was isolated—the rawsonol—which had an inhibitory effect in HMG-CoA reductase activity, as it was a rate-controlling enzyme of the mevalonate pathway that produces cholesterol molecules [149]. The study of Estrada et al. [150] identified other bromophenols, isolated from *Cymopolia barbata*, with antibacterial activity against *Staphylococcus aureus* and *Pseudomonas aeruginosa*; the compound was a brominated monoterpenoid quinol.

#### 4.1.2. Flavonoids

Other phenolic compounds present in green seaweeds, the flavonoids compounds, have been investigated from a medical perspective, such as the anti-diabetic area. Actually, a research on *Ulva prolifera* revealed that flavonoids-rich extracts under 3 KDa molecular weight promoted the decrease of fasting blood glucose, augmentation of oral glucose tolerance, and protection against liver and kidney injury with reduced inflammation in diabetic mice [151,152]. The mechanism of action also showed a modulation of the intestinal microbiome by the growth of the bacteria *Lachnospiraceae* sp. and *Alisties* sp. in overall abundance, which has a clear influence on the release of intestinal hormones that have a direct positive impact on insulin release and resistance [151].

*Caleurpa* spp. (Chlorophyta) have various flavonoids, such as kaempferol and quercetin. These flavonoids have been correlated with antioxidant activity [153]. *Caulerpa corynephora* has the same concentration of the rutin hydrate as the brown seaweed *Undaria pinnatifida* [35].

### 4.2. Red Seaweeds

The ecological function of the phenolic compounds in red seaweeds (Figure 11) has been barely investigated, but they probably have multipurpose actions in cell life, as antioxidants, chelating and defense against herbivory agents, as well as cofactors or hormones [154]. However, there is not always information on the bioactivity of the phenolics isolated, because some assays do not occur with purified groups of phenolic compounds but with an extract enriched in polyphenolics [154].

For instance, Farideh Namvar and colleagues [155] studied the effect of *Kappaphycus alvarezii* polyphenol-rich extract (ECMES) on cancer cell lines. The concentrations applied in this study did not demonstrate a cytotoxic effect on the normal cells, though it was cytotoxic to the MCF-7 cancer cell line. These results suggest that the ECME’s active substance might target cancer-associated receptors, cancer cell signaling molecules, or gene expression of the MCF-7 breast cancer cells that triggers mechanisms causing cancer cell death [155].

Phenols that are applied in food and drugs have been obtained from red seaweeds; the advantages of their bioactivities will be discussed further in Section 5 of this review.

The red seaweed has been characterized with different phenolic compounds, such bromophenols, flavonoids, phenolic terpenoids, and mycosporine-like amino acid.

#### 4.2.1. Bromophenols

From red seaweeds, phenolic compounds identified as bromophenols and benzoic acids have been the most researched, from isolation to conducting an extensive characterization [154]. Bromophenols are phenolic compounds that are prominently found in red seaweeds, with bromine substituent indistinct degrees [33].

Bromophenols isolated from *Symphyocladia latiuscula* have antioxidant activity against the DPPH assay. These phenolic compounds have various and diverse high brominated groups mainly based in the 3,4-dihydroxy-2,5,6-tribromobenzyloxy [33,102,156]. The same occurred with the *Polysiphonia stricta* (formerly known as *Polysiphonia urceolata*), revealing that the antioxidant power of red seaweeds depends on brominated units and degrees of brominating of the molecules [33,102,156]. Isolated red seaweeds bromophenols are mainly studied in the oncology, diabetic, and microbial fields, to observe their properties.

One of the main research studies of the red seaweeds bromophenolic compounds is the study of the compounds of the oncology area, with a lot of studies with isolated compounds having demonstrated interesting characteristics, as we demonstrate in this topic.

The 3-bromo-4,5-dihydroxy benzoic acid methyl ester and 3-bromo-4,5-dihydroxy-benzaldehyde bromophenols isolated from *Rhodomela confervoides* demonstrated high potential against KB, Bel-7402 (Human papillomavirus-related endocervical adenocarcinoma), and A549 cancer cell lines [157].

In addition, the bis(2,3-dibromo-4,5-dihydroxybenzyl) ether has demonstrated apoptotic activity against K562 human myelogenous leukemia cells [76], as has the lanosol butenone (Colensolide A) isolated from *Vidalia colensoi* (formely *Osmundaria colensoi*), revealing selectivity against leukemia cells [77,157]. The bis(2,3-dibromo-4,5-dihydroxybenzyl) ether showed anti-angiogenesis effects in vitro and in vivo (zebrafish embryos) by reducing the HUVEC (human umbilical vein endothelial cells) cells’ proliferation, migration, and tube formation; however, it did not decrease the preformed vascular tube. Overall, it indicates a potential for further studies for cancer prevention and novel therapies, due to the unique structure that is different from the current anti-angiogenesis therapeutic agents [158].

The bis(2,3-dibromo-4,5-dihydroxybenzyl) methane isolated from *R. confervoides* demonstrated activity against the BEL-7402 (Human papillomavirus-related endocervical adenocarcinoma) cancer cell line, inhibiting the cell adhesion to fibronectin and collagen as well as cell migration and invasion, demonstrating an interesting anti-metastatic activity that can be developed to understand how it can be applied in oncologic therapeutic and compound selectivity [159]. From this seaweed, 8 bromophenols have been isolated that have been assayed against various cancer cell lines, such as HCT-8 (human colon cancer), Bel7402, BGC-823 (stomach cancer), A549 (adenocarcinomic human alveolar basal epithelial cell), and A2780 (human ovarian cancer), with only four compounds demonstrating activity against the cancer cell lines tested: 2,3-dibromo-4,5-dihydroxyphenylethanol, 2,3-dibromo-4,5-dihydroxyphenylethanol sulfate, 3-bromo-4,5-dihydroxyphenylethanol sulfate, and 3-bromo2-(2,3-dibromo-4,5-dihydroxybenzyl)-4,5-dihydroxyphenylethanol sulfate [160].

The bromophenols also show interesting results regarding their anti-diabetic and anti-obesity medical aspects. The bromophenols extracted from *S. latiuscula*, such as the 2,3,6-tribromo-4,5-dihydroxybenzyl methyl ether and its derivates, inhibit α-glucosidase, which improves insulin sensitivity and glucose uptake [161]. In this species, there were also various bromophenolic compounds isolated (2,2′,3,6,6′-pentabromo-3′,4,4′,5-tetrahydroxydibenzyl ether, bis(2,3,6-tribromo-4,5-dihydroxyphenyl)methane, and 2,2′,3,5′,6-pentabromo-3′,4,4′,5-tetrahydroxydiphenylmethane) that demonstrated aldose reductase inhibitory activity, via an enzyme in the polyol pathway, which is responsible for fructose formation from glucose and has an important role in the development of diabetes [162].

Bromophenol derivatives based on 2,3-dibromo-4,5-dihydroxybenzyl units and highly brominated isolated from *Rhodomela confervoides* demonstrate activity against PTP1B (Protein tyrosine phosphatase 1B), which is a negative regulator of the insulin signalling pathway, and the compounds demonstrated a substantial decrease of the blood glucose in diabetic rats [163].

Ko et al. [66] isolated the 5-bromo-3,4-dihydroxybenzaldehyde from *Polysiphonia morrowii* and studied the effect and mechanism of action of this compound in adipogenesis and the differentiation of 3T3-L1 preadipocytes, demonstrating that this bromophenol targets the peroxisome proliferator-activated receptor-γ expression levels and also the CCAAT/enhancer-binding proteins α and sterol regulatory element-binding protein 1. However, the main mechanism is under the AMP-activated protein kinase signal pathway, inhibiting the adipogenesis, in vitro assay. So, there is a need to develop further work to understand if the effect can be replicated in vivo.

The study of Mikami et al. [164] revealed that the n-butyl 2,3-dibromo-4,5-dihydroxybenzyl ether, isolated from the red seaweed *Odonthalia corymbifera*, demonstrated inhibition against the glucose 6-phosphate dehydrogenase, which is a key enzyme in the formation of reduced nicotinamide adenine dinucleotide phosphate (NADPH). This compound is an important factor in the biosynthesis of fatty acid and cholesterol and a factor intrinsic to cancer cells lines’ growth. These studies used bacterial and yeast glucose 6-phosphate dehydrogenase enzymes, so there is a long road to understand its effect in humans, but the assay has proven the bioactive potential of the bromophenolic compound.

Researchers have proven the antimicrobial action of these molecules. For instance, bromophenols isolated from *R. confervoides*, by Xu et al. [165], were tested against various bacteria species, such as *Staphylococcus aureus*, *Staphylococcus epidermidis*, *Escherichia coli,* and *Pseudomonas aeruginosa*. The results were the most promising against the *Staphylococcus* species, and the best bromophenol in the assay was bis (2,3-dibromo-4,5-dihydroxybenzyl) ether, without effect against only one *Escherichia coli* strain.

In addition, the lanosol methyl ether, lanosol butenone, and rhodomelol isolated from the *Vidalia colensoi* (formerly known as *Osmundaria colensoi*) demonstrated anti-bacterial and anti-fungal activities against various pathogens, such as various *Halomonas* species, *Pseudomonas* sp., *Vibrio alginolyticus*, *Vibrio harveyi*, *Klebsiella pneumoniae*, *Propionibacterium acnes*, *Staphylococcus aureus*, *Alternaria alternata,* and *Candida albicans*. These effects proved to be bactericidal and bacteriostatic or fungicidal and fungistatic, demonstrating the dose-dependent curve for effect against the pathogen. These results prove the multi-performance of the red seaweeds bromophenols against terrestrial and aquatic microbes [166].

The bis(2,3-dibromo-4,5-dihydroxybenzyl) ether demonstrated anti-fungal activity against various phytopathogenic fungi, such as *Botritys cinerea*, *Valsa mali*, *Fusarium graminearum*, *Coniothyrium diplodiella*, and *Colletotrichum gloeosporioides*, and it did not demonstrate activity against *Alternaria mali* and *Alternaria porri*, indicating that the research of bromophenols applications can pass also to control pathogens in the food and contribute to high food security [167].

The *S. latiuscula* bromophenol 2,3,6-tribromo-4,5-dihydroxybenzyl methyl ether had demonstrated anti-viral effect against various types of herpes simplex type 1. This assay was performed in mice with anti-viral activity at a dosage of 20 mg/kg for 6–10 days, without the presence of major secondary effects observed. The behavior of the anti-viral activity was comparable with acyclovir, in the skin lesions and the concentration of the virus particles in the brain [168].

The bromophenols isolated from *Polysiphonia morrowii*, the 3-bromo-4,5-dihydroxybenzyl methyl ether (BDME) and 3-bromo-4,5-dihydroxybenzaldehyde (BD), demonstrated anti-viral activity against the infectious hematopoietic necrosis virus (IHNV) and infectious pancreatic necrosis virus (IPNV), which are two aggressive fish pathogenic virus. The study of Kim et al. [169] demonstrated that the BDME has an effect against the two viruses; however, the BD only has an effect against IHNV. It ought to be noted that the anti-viral effect is lower than the control used (ribavirin).

These molecules, isolated from the red seaweeds, have other interesting activities. For example, 3,5-dibromo-4-hydroxyphenylethylamine, 2,2′,3,3′-tetrabromo-4,4′,5,5′-tetrahydroxydiphenylmethane, 2,3-dibromo-4,5-dihydroxybenzyl alcohol, 2,3-dibromo-4,5-dihydroxybenzyl methyl ether, 2,2′,3-tribromo-3′,4,4′,5-tetrahydroxy-6′-hydroxymethyldiphenylmethane, and 3-bromo-4-(2,3-dibromo-4,5-dihydroxybenzyl)-5-methoxymethylpyrocatechol, isolated from *Odonthalia corymbifera*, had been assayed against a fungal pathogen that affects the rice plants (*Magnaporthe grisea*). The bromophenols demonstrated efficiency in reducing the disease impact in the wild rice plants, with comparable results of rice genetic modified to be resistant to this pathogen [170].

In addition, 2,3,6-tribromo-4,5-dihydroxybenzyl alcohol and bis-(2,3,6-tribromo-4,5-dihydroxybenzyl) ether, extracted from *S. latiuscula*, act as multitarget ligands promoting neuroprotection [161].

#### 4.2.2. Flavonoids

The flavonoid isolation from *Acanthophora specifera* demonstrates a mixture of chlorogenic acid (69.64%), caffeic acid (12.86%), vitexin-rahmnose (12.35%), quercetin (1.41%), and catechol (0.59%) [171]. The flavonoid-enriched extract has demonstrated antioxidant activity [172].

The study of Saad et al. [173] demonstrated a flavonoid-enriched extract from *Alsidium corallinum* containing flavonoids such as luteolin 5,7,3′,4′ tetramethyl ether, quercetin 3,7 dimethylether 4′ sulfate, and catechin trimethyl ether, which can be useful tools against the kidney dysfunction provoked by potassium bromate; this assay was done in mice.

#### 4.2.3. Phenolic Terpenoids

Davyt et al. [174] isolated 11 sesquiterpenes from *Laurencia dendroidea* (formerly known as *Laurencia scoparia*) and studied the anthelmintic activity with moderate results against the parasite stage of *Nippostrongylus brasiliensis*. Meroditerpenes of the *Callophycus serratus* and *Amphiroa crassa* had been demonstrated in the initial stages of studying antimalarial activity [175].

A chromene-based molecule isolated from *Gracilaria opuntia* has been assayed and proved to have antioxidant and also anti-inflammatory activity in in vitro assays [176].

#### 4.2.4. Mycosporine-Like Amino Acid

This class of phenolic compounds is exclusive from red seaweeds and can be found in various species, such as *Asparagopsis armata*, *Chondrus crispus*, *Mastocarpus stellatus, Palmaria palmata*, *Gelidium* spp., *Pyropia* spp. (formerly known as *Porphyra* spp.), *Crassiphycus corneus* (formerly known as *Gracilaria cornea*), *Solieria chordalis*, *Grateloupia lanceola*, and *Curdiea racovitzae* (Rhodophyta). Various MAAs have already been isolated—likewise, the palythine, shinorine, asterina-330, porphyra-334, palythinol, and usujirene. These types of compounds have a high antioxidant and photoprotection activity, and also anti-proliferative activities in the HeLa cancer cell line (human cervical adenocarcinoma cell line) and HaCat (human immortalized keratinocyte). This was already passed from the studies to the final product production [177,178,179,180,181,182,183,184]. Recent studies indicate that MAAs can have other important bioactivities, such as anti-inflammatory, immunomodulatory, and wound-healing properties [185,186,187]. So, these compounds are a natural alternative to the synthetic UV-R filters in the sunscreens. Thus, MAAs seem to be driven and focused to a specific area that permitted studying and rapidly obtaining a product that can be applied in humans.

### 4.3. Brown Seaweeds

Brown seaweeds (Figure 12) also have a polyphenols group of high interest: the phlorotannins, which are found in neither green nor red seaweeds. As other phenols, phlorotannins have strong antioxidant effects, especially phloroglucinol, eckol, and dieckol, which can be extracted from *Ecklonia cava*. Moreover, these molecules revealed effectiveness in protecting DNA against hydrogen peroxide, which induces damage [188]. Other phenolic compounds isolated from brown seaweeds are bromophenols, flavonoids, and phenolic terpenoids, which have been the least studied due to the high quantity of phlorotannins.

Polyphenols obtained from *F. vesiculosus,* at concentrations that were not cytotoxic, inhibited both HIV-1-induced syncytium formation and HIV-1 reverse transcriptase enzyme activity [189]. HIV-1 is not the only virus that is susceptible to Phaeophyceae seaweeds’ phenolics; murine norovirus (MNV) and feline calicivirus (FCV) are other examples [190,191].

#### 4.3.1. Phlorotannins

Contrasting the recent developments in other phenolic compounds classes from seaweeds, early research of isolation and characterization was performed in phlorotannins accumulated by brown seaweeds. Phlorotannins are the most studied group of phenolic compounds from algae [45].

Their antioxidant power is 2 to 10 times higher when compared to ascorbic acid or tocopherol [192,193], demonstrating a hypothesis to treatments of inflammatory diseases [194]. Additionally, it was proven that phlorotannins can be applied as a protective agent against the toxicity of drug/antibiotics in humans, without the drugs losing their power, diminishing the damage from drug-based toxicity, such as gentamicin [195].

These research studies occur mainly in cell lines and model organisms, so in the future, the studies must be advanced to the next phase of trials in humans; with safeguards, they could be applied in commercial products.

One of the principal areas of study using phlorotannins is the oncology area. It was demonstrated that dioxinodehydroeckol, dieckol, and phlorofucofuroeckol, isolated from *Ecklonia cava,* have anti-proliferative, anti-tumor, anti-inflammatory, anti-adipogenic, and anti-tumorigenic activities. These properties are mainly against breast cancer cell lines MCF-7 and MDA-MB-231, SKOV-3 (ovarian cancer line), HeLa (cervical cancer cell line), HT1080 (fibrosarcoma cancer cell line), A549 (adenocarcinomic human alveolar basal epithelial cell), and HT-29 (human colon cancer cell line) [196,197,198,199,200]. The dieckol demonstrated a high anticancer activity against the non-small–cell lung cancer line A549 [201]. Eckol inhibited the proliferation of SW1990 pancreatic cells induced by Reg3A (a pancreatic inflammation upregulated protein with pro-growth function) [202]. Dieckol isolated from *E. stolonifera* induced apoptosis in human hepatocellular carcinoma Hep3B cells, demonstrating potential to be a new therapeutic agent to treat liver tumor, but more research is needed in order to develop a secure form to be effective and safe, consequently, full understanding of the mechanism of action that has reported by Yoon et al. is needed [203].

Other assays in radiotherapy observed the radio-protective effects of dieckol, triphlorethol-A, and eckol from *Ecklonia* genus to γ-irradiation; thus, the phlorotannins can have multiple roles in protection from radiation aggression or use in radiotherapy. Mainly, the mechanism of action is the radical scavenging and reducing pro-apoptotic molecules, but dieckol has an impact on the increase of the enzyme manganese superoxide dismutase that prevents DNA damage and lipid peroxidation, accelerating the hematopoietic recovery [204,205,206,207,208]. This radiation-induced protection has also been proven with eckol (*Ecklonia* genus) in the intestinal stem cells damaged by gamma irradiation, although the mechanism of this effect was not fully understood [204].

Several studies also reported the anti-diabetic properties of the phenolics of brown seaweeds [209,210]. *Ascophyllum nodosum* [36], *Fucus distichus* [211], and *Padina pavonica* [212] are species that produce phlorotannins, which were proven to have this effect. Other in vivo investigations, testing eckol and dieckol from *Ecklonia cava* [213,214], *Ecklonia stolonifera,* and *Eisenia bicyclis* (formerly known as *Ecklonia bicyclis*) [215], demonstrated the effects of oral administrations of these phenols in diabetic models in alleviating the postprandial hyperglycemia, suggesting a reduction in insulin resistance in those animals [214]. Octaphlorethol A isolated from *Ishige foliacea* has also shown anti-diabetic activity in type 2 diabetes. The mechanism of action studied by Lee et al. [216] elucidates the clinical applications of this phlorotannin as a new drug candidate for the treatment of type 2 diabetes.

The compound isolated from *Sargassum patens*, 2-(4-(3,5-dihydroxyphenoxy)-3,5-dihydroxyphenoxy) benzene-1,3,5-triol (DDBT), suppresses in high quantity the hydrolysis of the amylopectin by human salivary and pancreatic α; with these results, it can be beneficial as a natural nutraceutical to prevent diabetes [217].

From the study of Oh et al. [218], the dieckol isolated from *E. cava* attenuates the leptin resistance in the brain, and furthermore, it can cross the blood–brain barrier, which demonstrates dieckol as a potential treatment for obesity.

The 6,6′-bieckol compound isolated from *E. cava* demonstrated an inhibition for the high-glucose-induced cytotoxicity in human umbilical vein endothelial cells (HUVECs) and insulinoma cells, showing it can be a potential therapeutic agent against the hyperglycemia-induced oxidative stress. This problem results in diabetic endothelial dysfunction [219,220].

These molecules have also shown anti-microbial effects. Dieckols (8.4′′′-dieckol, 6.6″-dieckol and 8.8′-dieckol) extracted from the species *E. cava* are inhibitors of the human immunodeficiency virus (HIV-1) reverse transcriptase, with 8.8′-dieckol being the most effective against reverse transcriptase [221,222,223].

The dieckol (isolated from *E. cava*) can interfere in the viral replication mechanism of SARS-CoV, more concretely in SARS-CoV 3CL protease *trans/cis*-cleavage with a high association rate with a dose-dependent effect on 3CL protease hydrolysis [224]. Additionally, this compound demonstrates a potential against the influenza A virus neuraminidase, the most promising target, because it is a critical role for viral life cycle [225]; however, the phlorofucofuroeckol was also the best phlorotannin in the assay, equal to dieckol. Even further, these compounds have synergy with oseltamivir to enhance the inhibitory effects.

In recent docking studies by Gentile et al. [226] of the chemical structure of heptafuhalol A, 8,8′-bieckol, 6,6′-bieckol, and dieckol, compounds already isolated and characterized from *E. cava*, it was found that they are the most active inhibitors from the marine origin of the SARS-CoV-2 protease, revealing great potential to be further investigated against SARS-CoV-2.

Eckol, dieckol, 8,8’-bieckol, 6,6’-bieckol, and phlorofucofuroeckol-A from *E. bicyclis* demonstrated anti-viral activity against human papilloma virus [227].

Dieckol and phlorofucofuroeckol-A, both extracted from *E. bicyclis*, were demonstrated to have potent anti-viral action [190]. Moreover, phlorotannins extracted from the *Eisenia*/*Ecklonia* genera, such as eckol, dieckol, fucofuroeckol-A, 8,8′-Bieckol, and phlorofucofuroeckol-A, also exhibited anti-fungal activity [228] and anti-bacterial activity against Methicillin-resistant *Staphylococcus aureus* and other bacteria that are pathogenic not only in humans but also in plants [229,230,231].

According to several studies, these compounds also have potential to be applied in the treatment of bone diseases. Arthritis is one of the most prevalent chronic diseases; it is commonly characterized by the degradation of the cartilage matrix in bones joints. The dieckol and 1-(3′,5′-dihydroxyphenoxy)-7-(2″,4″,6″-trihydroxyphenoxy) 2,4,9-trihydroxydibenzo-1,4-dioxin (isolated from *E. cava*) demonstrated a reduction of the inflammatory response and cell differentiation in an in vitro assay, demonstrating potential to minimize the impact of arthritis symptoms [232].

Phlorofucofuroeckol A isolated from *E. cava* promotes osteoblastogenesis, which can be used in bone remodelling and reduce osteoporosis-related complication, mainly age-related bone disorders [233].

Other than these, dieckol extracted from *E. cava* demonstrated an interesting bioactivity related to chronic diseases, because it demonstrated anti-neuroinflammatory properties that can be essential to auto-inflammatory diseases [234]. This was also proven with dieckol extracted from *E. stolonifera* [235], demonstrating potential for therapeutic application in hepatotoxicity. Still, more studies are needed to develop this area. However, in the study of Lee and Jun [236], the 8,8′-bieckol from *E. cava* demonstrated a high inhibitory effect against β-Secretase and acetylcholinesterase, which are the principal factors for the development of the Alzheimer’s disease, with interesting docking assays to advance this compound even further as a drug candidate for the therapeutics of Alzheimer′s disease. Furthermore, Wang et al. [237] demonstrated in vitro the potential of eckmaxol, isolated from *E. maxima*, as a therapeutic multi-action mechanism to treat the neurotoxicity that develops in the Alzheimer’s disease.

The study of Seong et al. [238] focused on the action of phlorotannins in the mechanisms of anti-depressant and anti-Parkinson’s disease. Dieckol and phlorofucofuroeckol-A (isolated from *Ecklonia stolonifera*) demonstrated high inhibition against the monoamine oxidases and demonstrated full agonists with high potency at the dopamine 3 and 4 receptors, proving that these compounds can have a multi-role intervention in the potential treatment of psychological disorders and Parkinson’s disease.

These molecules have other potential biomedical properties. Dioxinodehydroeckol and phlorofucofuroeckol A extracted from *E. stolonifera* present an anti-allergenic effect [239]. Phlorofucofuroeckol-B isolated from *Eisenia arborea* also demonstrates an anti-allergy effect with the inhibition of histamine release [111]. More importantly, dieckol from *E. cava* demonstrates suppressing the immunoglobulin E-mediated mast cell and the passive anaphylactic reaction, diminishing the type I allergic responses; however, it was needed a relatively high dosage to suppress the hypersensitivity, so more studies are needed to fully understand it [240].

Lee et al. [241] demonstrated the suppression mechanism of action of dieckol (isolated from *E. cava*) in liver fibrosis, proving the potential of dieckol to be further evaluated to treat chronic liver inflammation provoked by alcohol abuse, metabolic diseases, viral hepatitis, cholestatic liver diseases, and autoimmune diseases.

Hypertension is one of the most common cardiovascular problems that origins/develops into other dangerous diseases in the world, where the angiotensin-converting enzyme (ACE) plays a major role in augmenting health risks. The work of Jung et al. [242] and Wijesinghe [243] demonstrated that phlorofucofuroeckol A (isolated from *E. stolonifera*) and dieckol (isolated from *E. cava*) have a high inhibition against the ACE in in vitro assays, ameliorating the hypertension symptoms and the risk of development of more life-risking treats. One explanation that was proposed is that the closed-ring dibenzo-1,4-dioxin moiety can be essential for the bioactivity as well as the molecule polymerization level [242].

Fucophlorethol C isolated from *Dactylosiphon bullosus* (formerly known as *Colpomenia bullosa*) has a lipoxygenase inhibition activity, in soybean, with comparable power to the nordihydroguaiaretic acid (obtained from Zygophyllaceae plants), which is the most well-known inhibitor. This inhibitor is expected to suppress various lipoxygenase-related diseases, such as psoriasis, asthma, rhinitis, and arthritis [244].

Phlorotannins can act as an anti-UVB protective agent; dioxinodehydroeckol (isolated from *E.* cava) demonstrated a protection on the HaCat cells, reducing the apoptosis provoked by UVB [245]. Additionally, phlorotannins are being researched as whitening and/or anti-wrinkling agents for cosmeceuticals, such as dieckol, dioxinodehydroeckol, eckol, eckstolonol, phlorofucofuroeckol A, and 7-phloroeckol (isolated from various brown seaweeds). These have shown promising inhibition of tyrosinase and hyaluronidase [5,246,247,248,249,250,251]. In addition, the 7-phloroeckol (isolated from *E.* cava) has proven that it can be a hair growth promoter agent by the study of Bak et al. [252].

Although more research needs to be done, these phlorotannins can be a key for natural biomarkers for various applications [253]. Hydroxytrifuhalol A, 7-hydroxyeckol, diphloroethol, fucophloroethol, and dioxinodehydroeckol can be applied as food biomarkers, because after intake, they can be detected by plasma or urine samples. Having a short half-life, they are considered a good short-term biomarker.

The molecules could also be applied in the control of the algal blooms, as the phlorofucofuroeckol isolated from *Ecklonia kurome* demonstrated a high algicidal activity comparable to epigallocatechin [254]. In the swine’s diseases, the dieckol, 7-phloroeckol, phlorofucofuroeckol, and eckol isolated from *E. cava* demonstrated high activity against porcine epidemic diarrhoea virus (PEDV), but it is not known whether the compound interacts in the viral entry or in viral replication [255].

#### 4.3.2. Bromophenols

In terms of brown seaweeds’ bromophenols compounds isolated and their respective bioactivity analysis, the information is generally weak, having less research published, and being mainly about the brown seaweed *Leathesia marina* (formerly known as *Leathesia nana*) [33]. Various bromophenols identified in red seaweeds also appear in brown seaweeds, such as bis-(2,3-dibromo-4,5-dihydroxy-phenyl)-methane [256].

The studies of Xu et al. [257] and Shi et al. [258] isolated various bromophenol compounds from *L. marina* and conducted assays against various cancer lines *in vitro* and in mice. Some of the cancer lines were A549 (lung adenocarcinoma), BGC-823 (stomach cancer), MCF-7 (breast cancer), Bel7402 (hepatoma), and HCT-8 (human colon cancer), with five of the compounds showing cytotoxic effects against various cancer cell lines. For example, 6-(2,3-Dibromo-4,5-dihydroxybenzyl)-2,3-dibromo-4,5-dihydroxybenzyl methyl ether, bis(2,3-dibromo-4,5-dihydroxybenzyl) ether, and 3-bromo-4-(2,3-dibromo-4,5-dihydroxybenzyl)-5-methoxymethylpyrocatechol presented the best results in anti-tumor activity. Additionally, the bromophenols from *L. marina* demonstrated a growth inhibition of Sarcoma 180 tumors in vivo (mice), demonstrating a potential for further investigation.

On the other hand, (+)-3-(2,3-dibromo-4,5-dihydroxy-phenyl)-4-bromo-5,6-dihydroxy-1,3-dihydroiso-benzofuran, isolated from *L. marina,* demonstrated an inhibitory activity against thrombin in in vitro and in vivo assays [70,259]. The compound reveals a potential to treat cardiovascular diseases, where thrombin can play a key role in the development of the diseases, such as thrombosis and thromboembolism, due to thrombin being a proteinase that plays a key role in the procoagulant factors.

#### 4.3.3. Flavonoids

The work of Agregán et al. [57] identified flavonoids in *Ascophyllum nodosum*, *Bifurcaria bifurcata,* and *Fucus vesiculosus*, which are mainly acacetin derivatives, gallocatechin derivatives, and hispidulin. This work demonstrates that the isolation work of the seaweeds’ flavonoids is only beginning, and some of the flavonoid’s structures are comparable with terrestrial plants, such as oregano (*Folium origani cretici*) or basil (*Ocimum basilicum*).

The myricetin isolated from *Turbinaria ornata* demonstrated attenuating the effect of rotenone-induced neuronal degeneration in *Drosophila melanogaster*. This result demonstrates that this flavonoid can have a positive impact on Parkinson’s disease in the muscular coordination and memory of the fly model used [260].

#### 4.3.4. Phenolic Terpenoids

Meroditerpenoids have been isolated and characterized from the *Treptacantha baccata* (formerly known as *Cystoseira baccata*), all of the compounds isolated have a bicycle (4.3.0) nonane ring; mainly, they have antifouling activity against algal settlements and mussel phenol oxidase, but they appear to be non-toxic and non-bactericidal for the larvae of sea urchins and oysters [261]. Nine tetraprenyltoluquinol-based meroterpenoids isolated from *Halidrys siliquosa* present antifouling properties and some have also demonstrated anti-bacterial activity [262].

Thus, meroditerpenoids isolated from *Stypopodium flabelliforme* have been assayed against the NCI-H460 (human lung cancer cell line) presenting a moderate response [263]. Meroditerpenoids, such as epitaondiol and stypodiol, isolated from *Stypopodium flabelliforme* demonstrated anti-bacterial activity against *Enterococcus faecalis*. These compounds demonstrated anticancer activity against Caco-2 (human epithelial colorectal adenocarcinoma), SH-SY5Y (neuroblastoma), and RBL-2H3 (Rat Basophilic Leukemia cells), and they especially affected the cancer line RAW.267 (Abelson murine leukemia virus-induced tumor), but the potential to be used as therapeutic is interesting, because they appear to be non-toxic against non-cancer cell line V79 [264]. It can be used as a selective anticancer drug, so there is more research to be done to fully understand the mechanism and how this unique characteristic can be explored. The epitaondiol from *Stypopodium zonale* has been assayed against human metapneumovirus with a prominent viricidal activity [265].

Zonarol, a sesquiterpene from *Dictyopteris undulata,* provides neuroprotection by protecting the neuronal cells from the oxidative stress, which is one of the main mechanisms of the phenolic compounds, so this is a candidate to be further studied to see if it is positively applicable as neurodegenerative diseases therapeutic [266].

Ali et al. [267] isolated plastoquinones, sargahydroquinoic acid, sargachromenol acid, and sargaquinoic acid from *Sargassum serratifolium* and demonstrated that these compounds have activity against protein tyrosine phosphatase 1B, so these compounds can be further explored for the prevention and treatment of type 2 diabetes.

## 5. Seaweed Phenolics: Commercial and Potential New Applications

The phenolic compounds that have been isolated from seaweeds are scarce, and further research will enlarge the biochemical library and improve the chance to discover new potential compounds to different industries or areas, so this area is still evolving along the road from isolation to application. This subtopic will describe the work done on the bioactivities described above and potential novel applications.

The major problem of these compounds to be inserted in real commercial applications is mainly the compound concentration in seaweed, due to the low extraction efficiency and seaweed biomass availability [15,45,105]. The solution goes through seaweed aquaculture that already exists but at a low level [268].

Although the seaweeds’ polyphenols are a target of ongoing research, the in vivo effects are scarce and unclear to their correspondence in vitro, which can be explained by the diversity of the methodology used [45].

Until now, the bioavailability of seaweeds has not been completely reinvestigated. More studies and research are needed in this field. Most of the seaweed phenolic pharmaceutical and biomedical bioavailability studies have been supported in mouse-model systems. Proof of the protecting effects of seaweed phenols against other diseases has been derived from animal experiments and in vitro studies. Consequently, new research studies are needed to examine and fully understand their bioavailability in humans (percentage of the substance that enters in the human circulation system and has an active effect) [45].

Phenolic compounds are the most researched seaweed compounds and are already applied in commercial solutions (e.g., cosmetic products). Normally, the phenolic compounds are not isolated, because the commercial products of seaweed extracts have a considerable quantity of phenols.

### 5.1. Food Applications

The antioxidant potential of these compounds could be seen as a natural and non-harmful food stabilizer and preservative, attracting interest in food industries. However, there is a need to take into account that oxidized phenolic compounds can react with amino acids to form insoluble complexes, which may inhibit proteolytic enzymes and thus decrease the nutritive values of a food product [269]. An investigation demonstrated a negative correlation between the phenolic content of the seaweeds *Ulva lactuca*, *Hypnea charoides,* and *Hypnea japonica* (phenolic content 38.8% ± 0.5%, 16.9% ± 1.0%, and 16.3% ± 0.03%, respectively) and the digestibility and amino acid bioavailability of 85.7% ± 1.9%, 88.7% ± 0.7%, and 88.9% ± 1.4%, respectively [269]. This is likely to be a greater issue in brown seaweed species, as they are typically higher in phenolic content, including catechins, flavonoids, and phlorotannins [19,270].

Nevertheless, the restrictions of the synthetic ingredients in the food industry can be the turning point for the exploitation of seaweed compounds as safe alternatives [45], as they also have anti-microbial activities against major food spoilage and food pathogenic microorganisms [271]. Extracts that have seaweed phenolic antioxidants have been applied as enhancers of the oxidative stability and to conserve or increase the intrinsic quality and nutritional value of foods [272,273].

Their antioxidant potential is useful in the food industry, not only as nutraceutical compounds for functional food products, in which they are of indubitable valuable for improving health (as food supplements), but also to extend the shelf-life period when applied in processed food (functional foods) [11,274]. Additionally, the antimicrobial potential from the seaweed phenolics demonstrates that they can be useful in the food industry [275].

Moreover, bromophenols from *Ulva lactuca* and *Pterocladiella capillacea* were studied as “marine flavor” agents in farmed fish and other aquatic organisms, because the farming final products can have different flavor from the wild catch, and this can be inserted as a feed ingredient or a seaweed bromophenol-enriched sauce [276].

Seapolynol^TM^ (Botamedi Inc, Seoul, Korea) is a food supplement that has been approved by the European Food Safety Authority [277]. This supplement is based on dieckol and other polyphenols from *E. cava*; it has been tested and showed promising results as an anti-hyperlipidemic [278] and cardioprotective agent against doxorubicin-induced cardiotoxicity [279]. Furthermore, Seapolynol^TM^ affected the sensitivity of insulin in type 2 diabetes and could play a key role in the prevention of metabolic disorders [280,281,282]. However, these assays were performed in mice.

### 5.2. Cosmetic Applications

The lack of toxicity of phlorotannins when compared with other natural antioxidants, and their effects in prevention the loss of skin elasticity of aged skin, is attracting the attention of researchers to develop novel natural cosmeceutical and pharmaceutical formulations [34,231,283].

Currently, some seaweed extracts containing the respective phenolic compounds, namely phlorotannins, are present in cosmetics, such as skin care and anti-aging products. Seaweeds are already used for this purpose; for instance, in *Saccharina japonica* (formerly known as *Laminaria japonica*), its extract (dasima extract produced Natural Solutions, in South Korea) is the ingredient utilized in facial masks, and it has anti-inflammatory, antioxidant, and anti-microbial effects. *Ecklonia cava* is an anti-inflammatory, antioxidant, anti-bacterial, and UV-protecting agent; furthermore, it is also anti-bacterial specifically against *Propionibacterium acnes*, which is a bacteria that causes acne. Thus, these phlorotannins are included in various cosmetic formulations as well [284,285,286,287].

The cosmetic industry is one of the main areas that drives the insertion of seaweed phenolic compounds in commercial applications, such as natural UV screening (Helioguard^®^ 365, produced by Mibelle Biochemistry in Buchs, Switzerland; Aethic Sôvée^®^ produced with Photamin, phenolic extracted from seaweeds, produced by AETHIC^®^ in London, UK from *Porphyra umbilicalis*) [95,288,289,290], or an anti-aging agent (ECKLEXT^®^ BG, produced by NOF Group, product obtained by phlorotannins enriched-extraction of *Ecklonia kurome*, harvested in Japan) [291]. An *Asparagopsis armata* extract containing MAAs is integrated in lotions with anti-aging properties [292].

Another part of cosmetics is the formulation of bioactive extracts for their incorporation in commercial formulas. The Natural Solution produces two registered extracts based on seaweeds that have phenolic compounds as active ingredients [293]:-*Ulva compressa* (formerly known as *Enteromorpha compressa)* Extract (Green Confertii Extract-NS by Natural Solution in Flemington, NJ, USA) contains flavonoids, tannins, polysaccharides, and acrylic acid as active compounds. It displays an antioxidant effect and anti-allergic effect and acts as an anti-microbial, antioxidant, and anti-allergic agent [294].-*Fucus vesiculosus* Extract (Bladderwrack Extract-NS by Natural Solution) contains fucoidan and phlorotannins as active compounds and acts as an anti-aging, antioxidant, anti-fungal, and anti-bacterial agent [295].

At last, there are new studies and developments in this area focusing on new phenolic compounds and cosmetic utilizations, with a view to patent the new compounds and formulations [296,297,298].

For this wide range of bioactivities and properties, the usage of natural anti-aging products derived from seaweeds is gaining reputation and attracting researchers’ consideration [34].

### 5.3. Pharmaceutical and Biomedical Applications

Seaweeds have been used for centuries as a normal medicine for diverse health diseases in folk medicine [299]. This attribute was considered as early as 300 BC in Asian cultures, and the Celtic, British, and Roman populations located near the sea used them for 1000 years for healing wounds, as vermifuge, or as anthelmintic, so the modern search in pharmaceutical and biomedical areas is evolving and in continuous progression [300,301].

Seaweed’s phenols have a wide range of possible applications in the pharmaceutical and biomedical areas due to their various bioactivities. Phenols, mostly the phlorotannins, have an outstanding antioxidant power due to their ability as chelating agents with reactive oxygen species and consequently preventing oxidative stress and cell damage [4,5]. Therefore, the scavenging of oxidants is important to control various diseases; thus, phenolic compounds of seaweeds are extremely valuable as a natural source of antioxidant agents.

Phenolic compounds are being researched for their application in human health to ameliorate or cure some of the main disease problems nowadays, such as cardiovascular, diabetes, neurodegenerative, and mental disorders [4,5].

Free radicals’ occurrence against macromolecules (e.g., membrane lipids, proteins, enzymes, DNA, and RNA) plays a pivotal role in several health disorders such as cancer, diabetes, neurodegenerative, and inflammatory diseases. Therefore, antioxidants may have a beneficial effect on human health by preventing free radical damage [33].

#### 5.3.1. Cardiovascular Disease

It was demonstrated that a flavonoid-enriched diet improves endothelial function and lowers the blood pressure [302]. Phlorotannins have a positive effect on the amelioration of hypertension [27]. A compound isolated from *Saragassum siliquastrum*, the sargachromenol D, demonstrated high potential to be a new drug for blood pressure control in severe hypertension in instances where it cannot be controlled by conventional combinatorial therapy [303].

Phlorotannins have been explored in the last decade by companies to obtain new products. The main targets of phlorotannin supplements in cardiovascular diseases are the arteriosclerosis prevention and the increase of protective high-density lipoprotein cholesterol (HDL-C). The products containing phlorotannins are HealSea^TM^ (produced by Diana Naturals in Rennes, France), IdAlg^TM^ (produced by Bio Serae in Bram, France), and Seanol^TM^ (produced by LiveChem in Jeju-do, South Korea and distributed by Simple Health, Maitland, USA) [45,304].

#### 5.3.2. Neurodegenerative and Mental Disorders

Some of the identified seaweed bromophenols can be included in schizophrenia and Parkinson’s disease therapies [161], because they can be used as a multitarget ligand to neurosensors. Furthermore, phlorotannins have neuroprotection action and may be the key to the treatment of neurodegenerative diseases [31].

Extracts of *Eisenia bicyclis* inhibited β-amyloid cleavage enzyme activity [305]. This potentiality was also demonstrated using phlorotannins of other brown seaweeds, such as *Ishige foliacea*, *Ecklonia maxima,* and *E. cava*, proving that they could be useful for Alzheimer’s disease treatment [31,234,237,306].

#### 5.3.3. Anticancer Properties

The phenolic compounds and their by-products can play significant roles as anticancer metabolites, performing in different parts of the evolution of cancer such as proliferative signalling, metastasis, cell cycle, resistance to cell death, evasion, angiogenesis, and the evasion of growth suppressors [8,307,308,309].

Chemotherapy is one of the main therapeutic approaches for cancer treatment, and there are already various natural anticancer molecules isolated—for example, the clinically used camptothecin and taxol [310].

The phenolic compound has a promising cytotoxicity against various cancer cells lines, and the selectivity of the compounds against cancer cell lines needs to be considered, as demonstrated in Section 4. This issue is one of the major problems for the current anticancer drugs [33]. Therefore, the phenolic compound can be cytotoxic to normal cell lines, such as the human embryo lung fibroblast (HELF), because the compounds can have low selectivity and can be counter-productive in their application, so a full batch of assays is needed to have a secure anticancer compound available. For this case, chemical modification can be needed to enhance the molecule selectivity [33,130,311], and more research will be necessary until a commercial product is released.

Ganesan and colleagues have proven the antioxidant effectiveness of *Euchema* sp., *Kappaphycus* sp., *Hydropuntia edulis* (formerly known as *Gracilaria edulis*), and *Acanthophora spicifera* extracts [312]. In this way, *Hypnea musciformis*, *H. valentiae*, and *Jania rubens* extracts were found to have this potential and others for carcinogenesis reduction and inflammatory diseases prevention [11].

#### 5.3.4. Diabetes

Bromophenols present anti-diabetic effects [33,161] that could be used in the research and development of a novel class of anti-diabetic drugs or in supplements and functional food products.

Red seaweeds’ bromophenols anti-diabetic effects were demonstrated by Matanjun and colleagues [313] using the species *Odonthalia corymbifera* and *Symphyocladia latiuscula*. More recently, this property was also confirmed in other Rhodophyta species, such as *Laurencia similis* [314], *Rhodomela confervoides* [163], and *Grateloupia elliptica* [315].

Polyphenols have these effects by inhibiting hepatic gluconeogenesis and reducing the activity of digestive enzymes such as α-amylase, α-glucosidase, lipase, and aldose reductase [125,210]. A commercial formulation of *A. nodosum* and *F. vesiculosus* phlorotannins, InSea2™ (Rimouski, QC, Canada), promotes a 90% reduction of the postprandial blood glucose, reducing the peak insulin secretion by 40% [316]. This leads these molecules to be considered as a potential natural alternative to the existing anti-diabetic drugs that have undesirable side effects [210]. Beyond their possible incorporation in pharmacological formulations, these molecules may also be seen as novel compositions for functional food or nutraceutical products.

#### 5.3.5. Anti-Microbial Function

The research for novel anti-microbial agents has been a “long-distance run” for many years, because the discovery of new effective drugs did not keep the pace of increasing anti-microbial resistance, especially bacteria. One problem is the lack and limitation of compounds in screening libraries [317,318]. This new research phase based in blue biotech can be fundamental to the improvement of these chemical libraries, and their extension with novel biomolecules. Initial studies had proven to be useful due to a real increment of reports of phenolic compounds with promising anti-microbial activities [33].

Purified phenolic extracts were found to have powerful antimicrobial effect against bacteria, fungi [37], and virus [221], revealing their potential for their use in pharmacotherapy.

In this field, there was not a clear passage from the assay to applications, but there is some patent work with phlorotannins-based anti-bacterial agents [319].

There is one patent comprising 6,6′-bieckol from *E. cava* for HIV-1 pharmaceutical composition, demonstrating interest to further explore the commercial use of phlorotannins from this species [320].

#### 5.3.6. Tissue and Bone Regeneration

Tissue and bone engineering is a new biomedical area where the phlorotannins are recently being tested [321]. This area focuses on the regeneration of damaged tissues, bones, or organs where biomedical scaffolds, cells, and growth factors are combined to obtain success [322]. The scaffolds used in regenerative applications need to provide biocompatibility, biodegradability, and appropriate mechanical properties for successful application [323,324,325].

A study by Im et al. [321] demonstrated that the insertion of phlorotannins on the fish collagen/alginate biocomposite obtained a better result in the cell proliferation. In the in vitro assay, this new mixture showed better results in calcium deposition and osteogenesis, demonstrating that this product with phlorotannins is a potential biomaterial for the bone tissue growth.

At the commercial level, Seanol^®^ (*E. cava* phenolic extract produced by Botamedi Inc., Seoul, Korea) was evaluated as an ingredient in hydrogels for bone regeneration with the ability to promote the anti-bacterial activity and to enhance the bone mineralization [326].

#### 5.3.7. Anti-Inflammatory

The symptoms of inflammation embrace the occurrence of inflammatory cells or mediators in the specific or non-specific location of tissue affected by adverse stimuli, as well as wounds, allergies, irritantions, or infections [327]. Various phenolics extracts and compounds already demonstrated the power to reduce inflammation—for example, *Carpodesmia tamariscifolia* (as *Cystoseira tamariscifolia*), *Treptacantha nodicaulis* (as *Cystoseira nodicaulis*), *Treptacantha usneoides* (as *Cystoseira usneoides*) and *Fucus spiralis* phlorotannin-purified extracts, and *Neoporphyra dentata* (as *Porphyra dentata*) flavonoid-enriched extract (containing catechol, rutin and hesperidin) [37,328]. Two different studies revealed that phlorotannins purified extracts were able to significantly reduce the levels of nitric oxide (NO) in mouse (RAW 264.7) macrophage cells previously exposed to a lipopolysaccharide from *Salmonella enterica* [37,328]. The investigation conducted by Katarzyna Kazłowska and colleagues demonstrated the anti-inflammatory effect of methanolic extract of the phenolic fraction of *N. dentata*, containing catechol, rutin, and hesperidin, which suppressed NO production via NF-kappaB-dependent iNOS (inducible nitric oxide synthase) gene transcription [328]. 

Another anti-inflammatory effect was observed in an *E. cava* phlorotannins extract through the arachidonic-dependent pathway by the downregulation of prostaglandin E2 [329].

In short, the anti-inflammatory mechanisms of action are correlated with the modulation of NO levels, which could be by direct scavenging and/or by decreasing NO production through the inflammatory signaling cascade or by inhibiting the enzymes involved in NO production [37,328,329].

#### 5.3.8. Other Medical Applications

Phlorotannin complements show sleep-promoting properties in mice as demonstrated in Section 4 of this review. In human trials, it prevents waking up after falling sleep in adults with sleep disturbances [330].

In a new assay, for caffeine sleep disruption problems, the effects of phlorotannin complements were analyzed against a sedative-hypnotic drug zolpidem (ZPD) in mice. The phlorotannin complement attenuated the effect of the caffeine sleep disruption effects, with identical results as the ZPD assay. Furthermore, phlorotannins did not change the delta activity during the non-rapid eye movement sleep, unlike ZPD, which decreased the delta activity, which is a negative symptom of taking ZPD. Thus, this study suggests that phlorotannins can be applied to relieve transitory insomnia symptoms [330].

In the treatment of wounds, there are not many studies, although the work of Park et al. [331] demonstrated the potential of phlorotannins extracted from *E. cava* to be incorporated in polyvinyl alcohol hydrogel for wound healing after application.

Additionally, investigations showed that dietary phenols are recognized as xenobiotics in humans, and although their absorption is very low in the small intestine (about 5–10%), they revealed prebiotic action [332]. Once they reach the gut, phenols may modify and produce variations in the microflora community by exhibiting prebiotic effects and antimicrobial action against pathogenic intestinal microflora [152,333,334].

### 5.4. Feed and Animal Health

Seaweed’s polyphenolic compounds have been demonstrated to be bioavailable for animals from the colon [335]. These phenolic compounds can be absorbed either directly in the upper digestive tract in untouched form or in the lower intestine after alteration by bacteria in the digestive tracts [336,337]. A study by Nagayama et al. [231] identified a possible application of phlorotannins extracted from *Ecklonia kurome* as anti-bacterial drugs that can be a natural substitute for the recently banned feed antibiotics.

Phlorotannins perform as prebiotics for both ruminant and monogastric animals while the dosage is given at low doses, under 5% in the animal diet. Phlorotannins might be involved in the feed of poultry and pigs only at low levels, commonly up to 5–6% in growing animals and never above 10% to prevent the over-dosage that causes negative effects in animal health. Using the recommended dosage, there is an escalation in productivity (increase in animal growth and milk production), quality (enhanced meat quality), and safety (decrease in shedding of pathogenic microorganisms; phlorotannins are more effective than the condensed tannins from terrestrial plant sources) of animal products due to phlorotannins. An amelioration of the immune system and rise in the anti-oxidative status of the animals were also observed [231,338,339,340,341,342,343,344,345].

*Ascophyllum nodosum* holds high amounts of phenolic compounds (mainly phlorotannins) that are insoluble in the digestive tract of animals [344,346]. Wang et al. [344] demonstrated that the *A. nodosum* extract contained a considerable concentration of phlorotannins (up to 500 g/mL), which reduced fermentation in mixed feed and barley grain feed in an in vitro assay. The effects were linear with the phlorotannins concentration. The *A. nodosum* based meal improved the slowly degraded protein fraction and protein degradability, and it was further beneficial to the feed digestibility when complementing low-quality feed diets [347].

Tasco^®^ (Dartmouth, Nova Scotia, Canada) from Acadian Seaplants is one of the *A. nodosum*-based feeds in the feed market [348], where various studies have been conducted to prove that phlorotannins enhance animal health [339,343,345,349].

### 5.5. Agriculture

The seaweed phenolic compounds that are identified in some commercial seaweed extracts can be a protection factor against plant diseases [350] because of their anti-microbial activity. Another potential role of polyphenols is the protection of plant organisms from damages caused by free radicals and other oxidants [336,351].

This field is in evolution, and normally, it is mainly explored using brown seaweeds because of the knowledge regarding phlorotannins. The commercialization of a solution with phenolic compounds as a stimulant for mycorrhizal and rhizobial symbiosis is already a patented work, which can happen as fertilization or treatment based mainly in the *Fucus* and *Ascophyllum* genus [352]. Another patent is based on liquid fertilizer with an active plant disease protection effect [353].

Some extracts produced by Maxicrop, Acadian Seaplants, and algae have a high content of humic-like polyphenols or polyphenols, which have been derived mainly from brown seaweeds in this type of industry [354]. One example is the *A. nodosum* analyzed from Norway and Nova Scotia that has 15–25% of extractable polyphenols with high molecular weight [355].

### 5.6. Other Applications

Phlorotannins can substitute bisphenol A (BPA), a very toxic substance, in the vinyl esters. The material designed and tested by Jaillet et al. [356] demonstrated a high thermal stability and thermomechanical properties, so this phlorotannins-based material can be applied as a substitute of BPA-based material in thermoset networks for composites. There is a long way to go before these compounds can be applied, but this demonstrates the potential of phlorotannins and the phenolic compounds of seaweeds.

## 6. Conclusions and Future Perspectives

The phenolic compounds discovered in seaweeds are an extensive and diverse group that are divided in four major classes and specific groups of terpenoids, where the predominant bioactivity of all is the anti-oxidative activity.

The phlorotannins represent the phenolic class in a more advanced stage of research with commercials products already available in diverse areas. However, the major compounds isolated are from the *Ecklonia* genus, indicating that research is approaching seaweeds with more biomass available, and so, the hypothesis of exploring these products commercially is more feasible. Therefore, the range of studies of phenolic compounds is wide, with the exception of only MAAs, which have focused on UV protection with large success. This bioactive compound can be applied in a wide range of industries such as food and feed, biomedical and pharmaceutical, agriculture, and electronics.

From the bibliography analyzed, this review demonstrates a good index of new developments in the pharmaceutical area and in other areas by using seaweed phenolics, but there is long road to understanding the major parts of the compounds already isolated, and there are other seaweeds that can be targeted for studies due to the easy seaweed cultivation.

We also demonstrate that the extractions and isolation methods are still being developed in order to be more ecological and intuitive to perform, with better quality, purity, and quantity of the phenolic compound extracted. So, the seaweed phenolics can be key players in the future in different areas that can help the practices of humankind be greener through being supported by natural compounds that were very difficult to obtain until the developments of the last decade.

## Figures and Tables

**Figure 1 marinedrugs-18-00384-f001:**
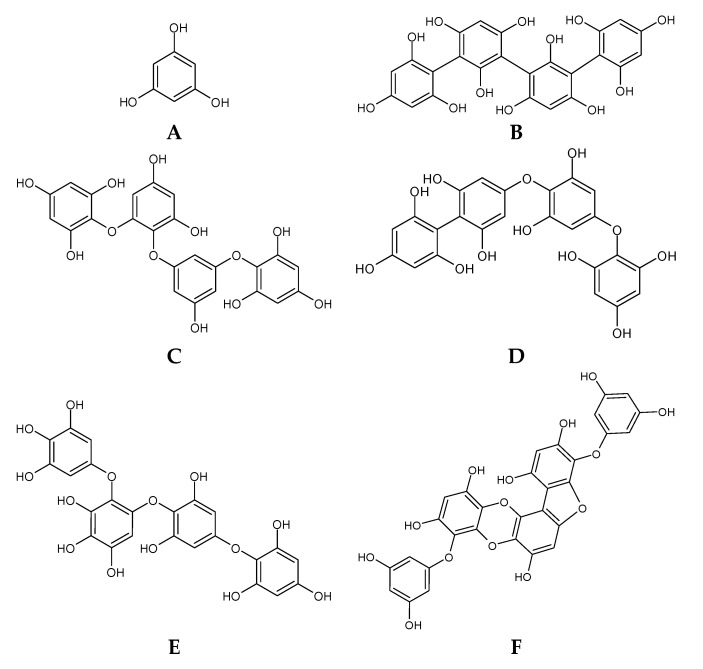
Chemical structures of phlorotannins: (**A**) Phloroglucinol; (**B**) Tetrafucol A; (**C**) Tetraphlorethol B; (**D**) Fucodiphlorethol A; (**E**) Tetrafuhalol A; and (**F**) Phlorofucofuroeckol.

**Figure 2 marinedrugs-18-00384-f002:**
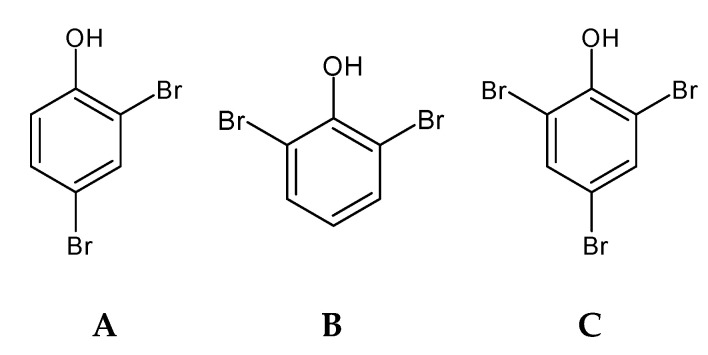
Chemical structures of bromophenols: (**A**) 2,4-bromophenol; (**B**) 2,6-bromophenol; (**C**) 2,4,6-tribromophenol.

**Figure 3 marinedrugs-18-00384-f003:**
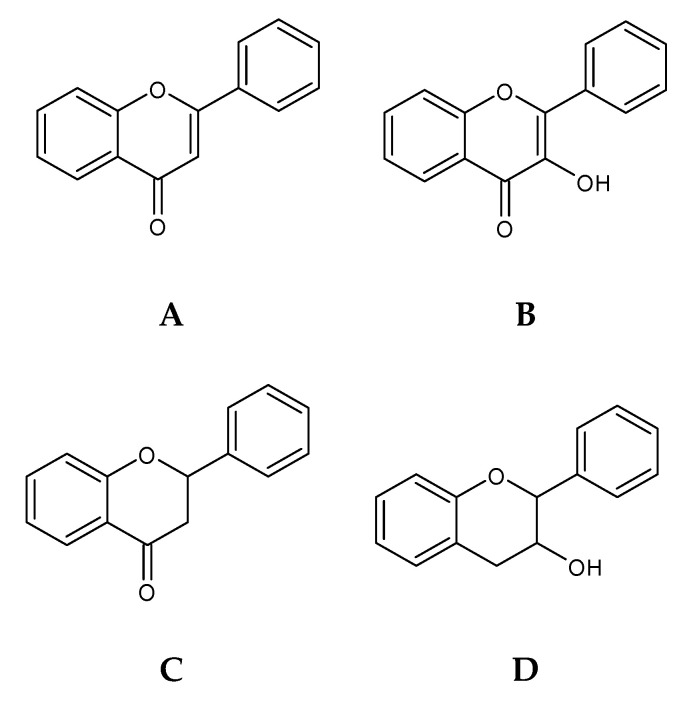
Main classes of flavonoids found in algae: (**A**) Flavones; (**B**) Flavonols; (**C**) Flavanones; (**D**) Flavan-3-ol.

**Figure 4 marinedrugs-18-00384-f004:**
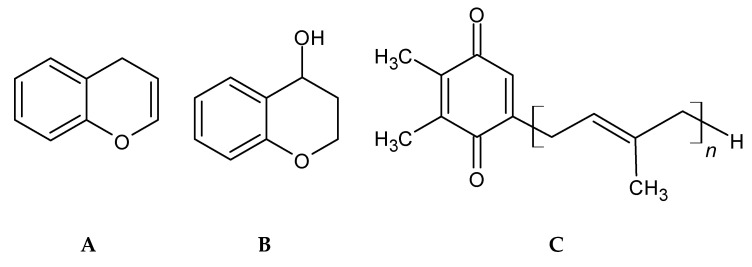
Main classes of phenolic terpenoids found in algae: (**A**) Chromene; (**B**) Chromanol; (**C**) Plastoquinone.

**Figure 5 marinedrugs-18-00384-f005:**
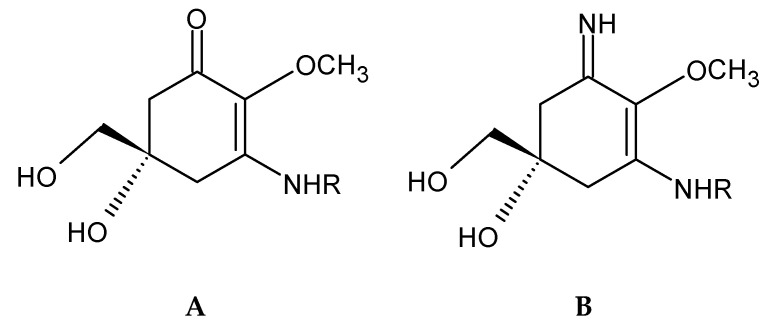
Mycosporine-like amino acids (MAA); (**A**) Aminocyclohexenone; (**B**) Aminocyclohexeniminone.

**Figure 6 marinedrugs-18-00384-f006:**
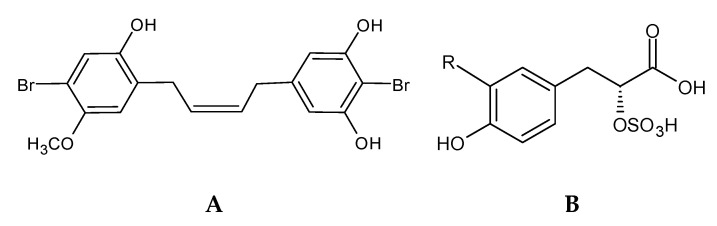
Other phenolic compounds in algae: (**A**) Colpol; (**B**) Tichocarpol.

**Figure 7 marinedrugs-18-00384-f007:**
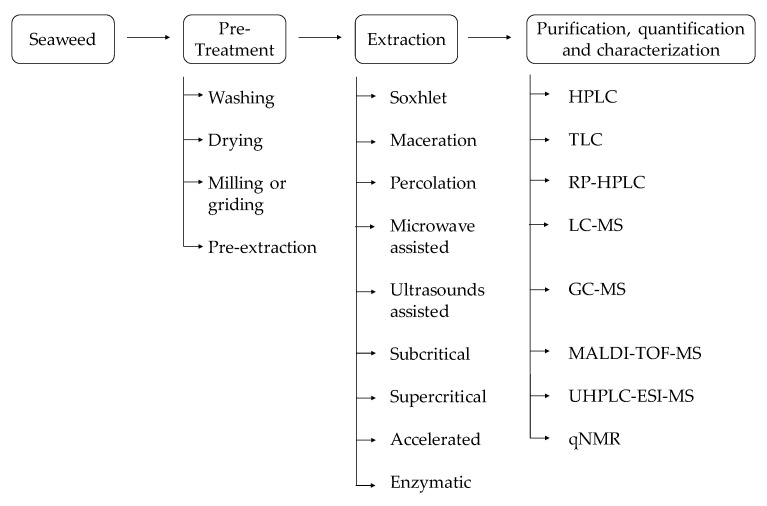
Schematic representation of possible methodologies for seaweed phenolic compounds quest.

**Figure 8 marinedrugs-18-00384-f008:**
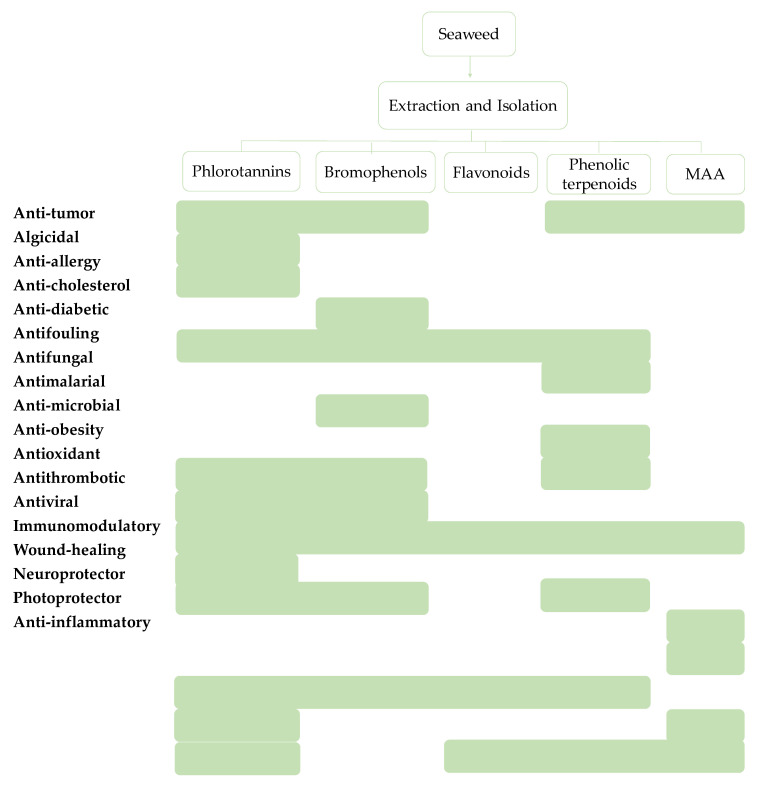
Biological activities from the different seaweed phenolic compounds reported in the literature.

**Figure 9 marinedrugs-18-00384-f009:**
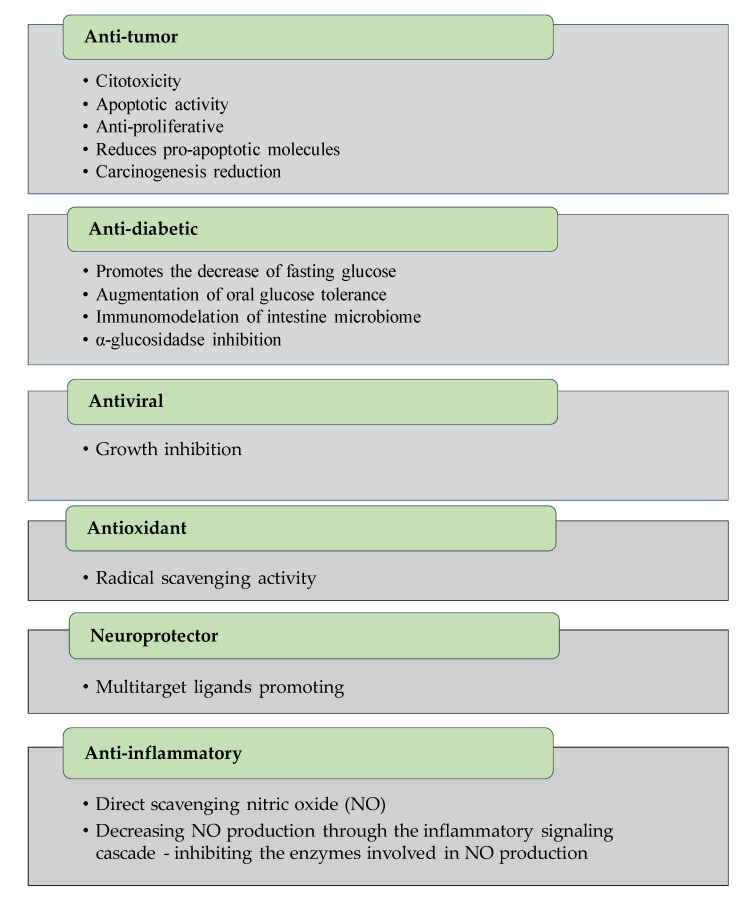
Seaweed phenolic compounds’ mechanisms of action.

**Figure 10 marinedrugs-18-00384-f010:**
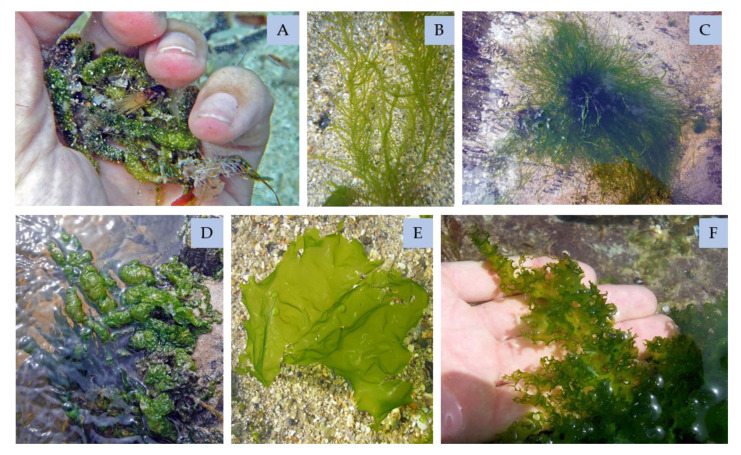
Examples of green seaweeds (Chlorophyta): (**A**)—*Dasycladus vermicularis*; (**B**)—*Ulva clathrata*; (**C**)—*Ulva compressa*; (**D**)—*Ulva intestinalis*; (**E**)—*Ulva lactuca*; (**F**)—*Ulva linza* [148].

**Figure 11 marinedrugs-18-00384-f011:**
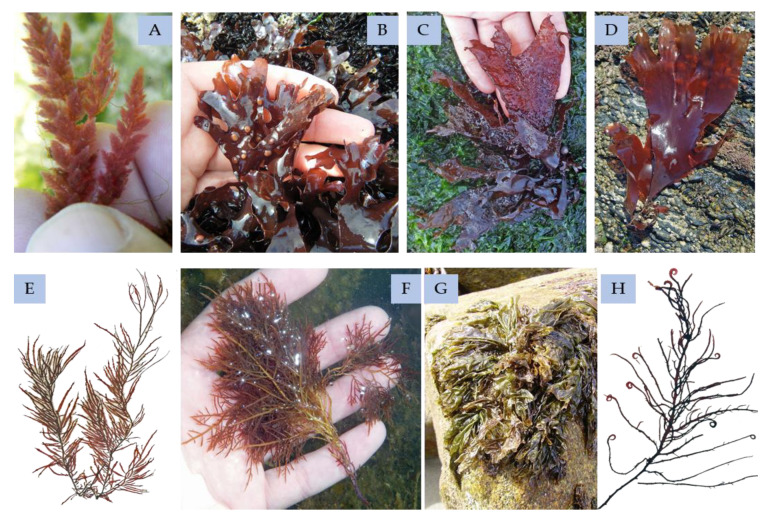
Examples of red seaweeds (Rhodophyta): (**A**)—*Asparagopsis armata*; (**B**)—*Chondrus cispus*; (**C**)–*Mastocarpus stellatus*; (**D**)—*Palmaria palmata*; (**E**)—*Solieria chordalis*; (**F**)—*Pterocladiella capillacea*; (**G**)—*Porphyra umbilicalis*; (**H**)—*Hypnea musciformis* [148].

**Figure 12 marinedrugs-18-00384-f012:**
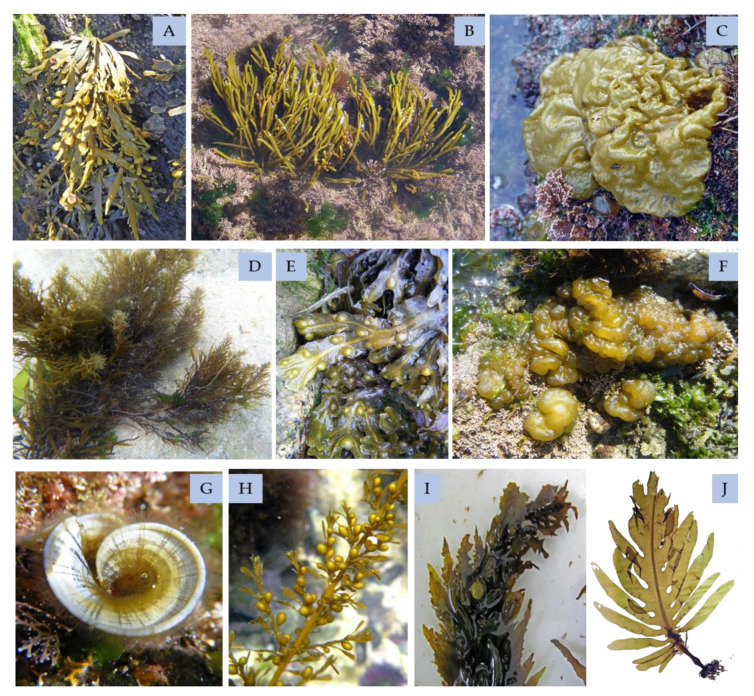
Examples of brown seaweed (Phaeophyceae): (**A**)—*Ascophyllum nodosum*; (**B**)—*Bifurcaria bifurcata*; (**C**)—*Colpomenia sinuosa*; (**D**)—*Treptacantha baccata*; (**E**)—*Fucus vesiculosus*; (**F**)—*Leathesia marina*; (**G**)—*Padina pavonica*; (**H**)—*Sargassum muticum*; (**I**)—*Sargassum vulgare*; (**J**)—*Undaria pinnatifida* [148].

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
