# Peer review of "Seaweed Phenolics: From Extraction to Applications"

_marinedrugs, 2020, doi:10.3390/md18080384_

Round 1

Reviewer 1 Report

In this review, Cotas and colleagues explain the importance that seaweed phenolics are demonstrating. Therefore, the authors have summarised the main groups of phenolic compounds and in which algae they are found, the main methods of extraction and purification, as well as their bioactivity and potential applications.

The main idea is interesting and this reviewer considers that it has been carried out correctly, writing a comprehensive summary, well documented and with relevant aspects. The references used are very recent (the authors even cite recent articles in which some of these compounds are capable of inhibiting SARS CoV-2 protease) and specify the main limitations that currently exist in this field of research.

The authors have included photographs of different algae that enrich the text.

I consider that it is a good review for this journal and that it would be publishable with small changes:

Major concerns:

  1. Due to all the information explained in the article, a couple of graphic summaries would be necessary. These images could specify the compounds explained and their main beneficial effects, the most important applications and a general diagram of the different forms of extraction and purification.
  2. How is the bibliography indicated in the text? It seems that there is no order or I have not understood how the bibliography is ordered.
  3. What can the authors say about phenolic acids? Why haven't they explained hydroxycinnamic acids and hydroxybenzoic acids in a separate section because of its importance?
  4. The section in which the researchers summarize the effects on inflammation seems to me too short. It would be interesting if the authors could specify on which cells and cytokines these compounds can act.
  5. Many of these compounds can act as prebiotics (the authors indicate this briefly). What beneficial effects have seaweed phenolics demonstrated on dysbiosis?

Minor concerns:

  1. Line 2: in the title, remove the "s" from the plural of “seaweed”.
  2. Line 46: include an "s" from the plural in “seaweed”.
  3. Line 54: include an "s" in “group”.
  4. Line 62: include an "s" in “seaweed”.
  5. Line 65: include an "s" in “seaweed”.
  6. Lines 70-73: I don't quite understand that paragraph. Aren't the sentences contradictory? Perhaps, a "however" in the last sentence will give more meaning to the paragraph.
  7. Line 164: The numbering is not correct. It should be 2.2. Bromophenols.
  8. Line 226: catechins
  9. Line 227: and
  10. Line 251: include an "s" in “seaweed”.
  11. Lines 377 and 378: the meaning is redundant. The phrase says the same thing as line 368.
  12. 12: Line 883: remove the "s" from  “cosmetic”.
  13. 13: Lines 901-904 say information very similar to lines 905-908.  Write the information for both of them in a single paragraph.
  14. 14: Line 1126: After “bisphenol A” you should write the acronym (BPA)

Author Response

Reviewer 1:

Comment 1: In this review, Cotas and colleagues explain the importance that seaweed phenolics are demonstrating. Therefore, the authors have summarized the main groups of phenolic compounds and in which algae they are found, the main methods of extraction and purification, as well as their bioactivity and potential applications.

The main idea is interesting, and this reviewer considers that it has been carried out correctly, writing a comprehensive summary, well documented and with relevant aspects. The references used are very recent (the authors even cite recent articles in which some of these compounds are capable of inhibiting SARS CoV-2 protease) and specify the main limitations that currently exist in this field of research.

The authors have included photographs of different algae that enrich the text.

I consider that it is a good review for this journal and that it would be publishable with small changes.

Answer 1: Firstly, we would like to thank the reviewer for his/her words. They were very valuable in improving the overall quality of the manuscript. Every aspect was addressed.

Comment 2: Due to all the information explained in the article, a couple of graphic summaries would be necessary. These images could specify the compounds explained and their main beneficial effects, the most important applications and a general diagram of the different forms of extraction and purification.

Answer 2: Thank you for your advice, we addressed the question. We added three graphical summaries.

Comment 3: How is the bibliography indicated in the text? It seems that there is no order, or I have not understood how the bibliography is ordered.

Answer 3: We have reviewed all references and we did as the Marine Drug as recommended.

Comment 4: What can the authors say about phenolic acids? Why haven't they explained hydroxycinnamic acids and hydroxybenzoic acids in a separate section because of its importance?

Answer 4: Thank you for your alert, we addressed the question in the section 2.1.

Comment 5: The section in which the researchers summarize the effects on inflammation seems to me too short. It would be interesting if the authors could specify on which cells and cytokines these compounds can act.

Answer 5: We added more information and references in the section 5.3.6.

Comment 6: Many of these compounds can act as prebiotics (the authors indicate this briefly). What beneficial effects have seaweed phenolics demonstrated on dysbiosis?

Answer 6: We added more information and references in the section 5.3.7.

Comment 7: Minor corrections:

  1. Line 2: in the title, remove the "s" from the plural of “seaweed”.
  2. Line 46: include an "s" from the plural in “seaweed”.
  3. Line 54: include an "s" in “group”.
  4. Line 62: include an "s" in “seaweed”.
  5. Line 65: include an "s" in “seaweed”.
  6. Lines 70-73: I don't quite understand that paragraph. Aren't the sentences contradictory? Perhaps, a "however" in the last sentence will give more meaning to the paragraph.
  7. Line 164: The numbering is not correct. It should be 2.2. Bromophenols.
  8. Line 226: catechins
  9. Line 227: and
  10. Line 251: include an "s" in “seaweed”.
  11. Lines 377 and 378: the meaning is redundant. The phrase says the same thing as line 368.
  12. Line 883: remove the "s" from “cosmetic”.
  13. Lines 901-904 say information very similar to lines 905-908.  Write the information for both of them in a single paragraph.
  14. Line 1126: After “bisphenol A” you should write the acronym (BPA)

Answer 7: We addressed the problem and all the text was trimmed and rewritten. All suggestions have been addressed.

Reviewer 2 Report

This is a very exhaustive revision about the Seaweeds phenolics that will be of potential interest to the readers. Only minor changes are suggested.

-In the Introduction section, there are some repetitions of concepts, and these should be revised and avoided. Also, some typographical errors can be found.

- Maybe, in the parragraph 3.2 Extraction, the authors could select the most suitable procedure for obtaining the best yield of bioactive compounds, based in the subsequent information provided in their revision

- When describing the biological activities of the different class compounds in the several types of seaweeds, it would be interesting to prepare some tables that would help to localize the compound and the corresponding biological activity (and the associated references), which would also allow to resume the text (sometimes difficult to read and follow).

-The section "5. Seaweed phenolics: commercial and potential new applications" should be revised to avoid repeated information already described in previous text in the manuscript. Maybe, the authors should focus on those applications not previously stated in the text.

- In addition, some figures explaining the potential mechanisms involved in the posible applications would be welcome

Author Response

Reviewer 2:

Comment 1: This is a very exhaustive revision about the Seaweeds phenolics that will be of potential interest to the readers. Only minor changes are suggested.

Answer 1: We would like to thank the reviewer words. We pleased your feedback. We addressed all the reviewer suggestions.

Comment 2: In the Introduction section, there are some repetitions of concepts, and these should be revised and avoided. Also, some typographical errors can be found.

Answer 2: We revised the section and trimmed and corrected the introduction section.

Comment 3: Maybe, in the paragraph 3.2 Extraction, the authors could select the most suitable procedure for obtaining the best yield of bioactive compounds, based in the subsequent information provided in their revision

Answer 3: We added more information in the topic.

Comment 4: When describing the biological activities of the different class compounds in the several types of seaweeds, it would be interesting to prepare some tables that would help to localize the compound and the corresponding biological activity (and the associated references), which would also allow to resume the text (sometimes difficult to read and follow).

Answer 4: We tried to added tables in section 4, however, we think that the data information was lost during the transformation of text plus table. Due to the majority of the text, we try to explain the compound mechanism of action (with compound isolated and respective bioactivity), even identical phenolic molecules can have different mechanism of action and different bioactivities. Consequently, the tables bring a lot of entropy in the text and did not shorten the text, due to the different assays’ types and mechanisms of action. Although, we thank your feedback and we created and added two figures in the beginning of the section 4, as graphical summaries of the section 4.

Comment 5: The section "5. Seaweed phenolics: commercial and potential new applications" should be revised to avoid repeated information already described in previous text in the manuscript. Maybe, the authors should focus on those applications not previously stated in the text.

Answer 5: We revised the section and trimmed the information.

Comment 6: In addition, some figures explaining the potential mechanisms involved in the possible applications would be welcome.

Answer 6: We added a general figure in the section 4 with more certain mechanisms of action. Because the phenols mechanism of action is very discussed without a totally agreement between the research studies analyzed.

Reviewer 3 Report

This manuscript makes a detailed review about seaweed phenolics. Some comments are listed below:

Line 42: “xanthopylls” instead of “xanthophyll’s”?

Line 210: add a point after “et al”.

Line 219: add a point after “et al”.

Figure 6 is not cited in the text.

Line 297. This sentence has also been menctioned in the introduction and topic 3 is not related with bioactivity. Therefore, it could be removed.

Line 306: “freeze-dried” instead of “freeze-drying”.

Lines 428-432. There is no reference to LC-MS analyses in seaweed phenolics and this is a common technique nowadays for polyphenols identification. Add some information about.

Line 448-449: “The relation between bioactivity and specific compound can be correlated”. This sentence is not well.

The term “bioavailability” has two different meanings in line 871 and 878 and it is a bit confusing. Clarify it, if posible.

Author Response

Reviewer 3:

Comment 1: This manuscript makes a detailed review about seaweed phenolics. Some comments are listed below:

 Line 42: “xanthopylls” instead of “xanthophyll’s”?

Line 210: add a point after “et al”.

Line 219: add a point after “et al”.

Figure 6 is not cited in the text.

Line 297. This sentence has also been mentioned in the introduction and topic 3 is not related with bioactivity. Therefore, it could be removed.

Line 306: “freeze-dried” instead of “freeze-drying”.

Lines 428-432. There is no reference to LC-MS analyses in seaweed phenolics and this is a common technique nowadays for polyphenols identification. Add some information about.

Line 448-449: “The relation between bioactivity and specific compound can be correlated”. This sentence is not well.

The term “bioavailability” has two different meanings in line 871 and 878 and it is a bit confusing. Clarify it, if possible.

Answer 1: We would like to thank the reviewer words. We pleased your feedback. We addressed the problem and all the text was trimmed and rewritten. All suggestions have been addressed.